# An oomycete effector subverts host vesicle trafficking to channel starvation-induced autophagy to the pathogen interface

Pooja Pandey[1†], Alexandre Y Leary[1†], Yasin Tumtas[1], Zachary Savage[1], Bayantes Dagvadorj[1], Cian Duggan[1], Enoch LH Yuen[1], Nattapong Sanguankiattichai[1], Emily Tan[1], Virendrasinh Khandare[1], Amber J Connerton[1], Temur Yunusov[2], Mathias Madalinski[3], Federico Gabriel Mirkin[1,2,3,4], Sebastian Schornack[2], Yasin Dagdas[3], Sophien Kamoun[5], Tolga O Bozkurt[1]*

[1]Imperial College London, London, United Kingdom; [2]Sainsbury Laboratory Cambridge University (SLCU), Cambridge, United Kingdom; [3]Gregor Mendel Institute (GMI), Austrian Academy of Sciences, Vienna BioCenter (VBC), Vienna, Austria; [4]INGEBI-CONICET, Ciudad Autonoma de Buenos Aires, Buenos Aires, Argentina; [5]The Sainsbury Laboratory, University of East Anglia, Norwich, United Kingdom

*For correspondence:
o.bozkurt@imperial.ac.uk

†These authors contributed equally to this work

Competing interests: The authors declare that no competing interests exist.

**Abstract** Eukaryotic cells deploy autophagy to eliminate invading microbes. In turn, pathogens have evolved effector proteins to counteract antimicrobial autophagy. How adapted pathogens co-opt autophagy for their own benefit is poorly understood. The Irish famine pathogen *Phytophthora infestans* secretes the effector protein PexRD54 that selectively activates an unknown plant autophagy pathway that antagonizes antimicrobial autophagy at the pathogen interface. Here, we show that PexRD54 induces autophagosome formation by bridging vesicles decorated by the small GTPase Rab8a with autophagic compartments labeled by the core autophagy protein ATG8CL. Rab8a is required for pathogen-triggered and starvation-induced but not antimicrobial autophagy, revealing specific trafficking pathways underpin selective autophagy. By subverting Rab8a-mediated vesicle trafficking, PexRD54 utilizes lipid droplets to facilitate biogenesis of autophagosomes diverted to pathogen feeding sites. Altogether, we show that PexRD54 mimics starvation-induced autophagy to subvert endomembrane trafficking at the host-pathogen interface, revealing how effectors bridge distinct host compartments to expedite colonization.

## Introduction

Autophagy is a conserved eukaryotic cellular process that mediates the lysosomal degradation and relocation of cellular cargoes within double-membraned vesicles called autophagosomes (*He and Klionsky, 2009*; *Leidal et al., 2020*). Although previously considered to be a bulk catabolic pathway tasked with maintaining cellular homeostasis under normal or stress conditions, it is now clear that autophagy can be highly selective (*Zaffagnini and Martens, 2016*). Autophagic cargoes are typically captured during autophagosome formation, a complex process regulated by a set of conserved autophagy-related proteins (ATG) as well as specialized autophagy adaptors and cargo receptors (*Mizushima et al., 2011*). These captured cargoes are sorted within the autophagosome during maturation of the isolation membrane (also known as the

**eLife digest** With its long filaments reaching deep inside its prey, the tiny fungi-like organism known as *Phytophthora infestans* has had a disproportionate impact on human history. Latching onto plants and feeding on their cells, it has caused large-scale starvation events such as the Irish or Highland potato famines.

Many specialized proteins allow the parasite to accomplish its feat. For instance, PexRD54 helps *P. infestans* hijack a cellular process known as autophagy. Healthy cells use this 'self-eating' mechanism to break down invaders or to recycle their components, for example when they require specific nutrients. The process is set in motion by various pathways of molecular events that result in specific sac-like 'vesicles' filled with cargo being transported to specialized compartments for recycling. PexRD54 can take over this mechanism by activating one of the plant autophagy pathways, directing cells to form autophagic vesicles that *Phytophthora* could then possibly use to feed on or to destroy antimicrobial components. How or why this is the case remains poorly understood.

To examine these questions, Pandey, Leary et al. used a combination of genetic and microscopy techniques and tracked how PexRD54 alters autophagy as *P. infestans* infects a tobacco-related plant. The results show that PexRD54 works by bridging two proteins: one is present on cellular vesicles filled with cargo, and the other on autophagic structures surrounding the parasite. This allows PexRD54 to direct the vesicles to the feeding sites of *P. infestans* so the parasite can potentially divert nutrients. Pandey, Leary et al. then went on to develop a molecule called the AIM peptide, which could block autophagy by mimicking part of PexRD54.

These results help to better grasp how a key disease affects crops, potentially leading to new ways to protect plants without the use of pesticides. They also shed light on autophagy: ultimately, a deeper understanding of this fundamental biological process could allow the development of plants which can adapt to changing environments.

phagophore) via the specific interactions between cargo receptors and ATG8, which decorates the phagophore to serve as a docking platform for cargo receptors (*Slobodkin and Elazar, 2013*; *Ryabovol and Minibayeva, 2016*).

The source of the phagophore is still under debate, but its primary source is considered to be the endoplasmic reticulum (ER) (*Bernard and Klionsky, 2013*; *Hamasaki et al., 2013*). As the cargo is captured, the phagophore undergoes massive expansion and is finally sealed to form a mature autophagosome. Therefore, formation of the autophagosome requires additional lipid supplies that are needed for elongation and final sealing of the phagophore. Supporting this view, the essential autophagy protein ATG2 was recently discovered to have lipid transfer activity (*Valverde et al., 2019*; *Maeda et al., 2019*; *Osawa et al., 2019*). To cope with cellular starvation, cells can rapidly generate hundreds of autophagosomes, conceivably requiring an available supply of lipids. Remarkably, in yeast, lipids mobilized from lipid droplets (LDs) were found to fuel autophagosome biogenesis during starvation-induced autophagy, which employs relatively larger autophagosomes (*Shpilka et al., 2015*). In contrast, smaller autophagosomes of the cytosol-to-vacuole targeting (Cvt) pathway do not rely on LDs, suggesting that LDs are specifically recruited for starvation-induced autophagy in order to meet the increased demand of lipids required for the biogenesis of larger sized autophagosomes (*Shpilka et al., 2015*; *Dupont et al., 2014*).

Although poorly characterized, there is accumulating evidence for autophagosome maturation relying on vesicle transport and membrane expansion events which are regulated by secretory pathways involving Rab GTPases (*Ao et al., 2014*). For instance, the mammalian Rab8a that regulates polarized secretion and lipid droplet fusion events is also implicated in autophagy (*Bansal et al., 2018*; *Vaibhava et al., 2012*). The model plant *Arabidopsis* genome contains five closely related Rab8a members organized into the RabE group (RabE1a-d), but whether any of the RabE1 members contribute to autophagy is unknown (*Rutherford and Moore, 2002*). Molecular mechanisms governing autophagosome biogenesis, including the sources of membrane precursors required for autophagosome elongation and the transport routes to position these lipid supplies at autophagosome assembly sites remain to be comprehensively described in plants.

The discovery of various autophagy cargo receptors uncovered a multitude of selective autophagy pathways implicated in crucial cellular functions ranging from development to immunity in both plants and animals (*Zaffagnini and Martens, 2016*). For instance, the plant selective autophagy cargo receptor Joka2/NBR1 mediates antiviral immunity by eliminating viral components (*Hafrén et al., 2017*; *Hafrén et al., 2018*; *Jung et al., 2020*). Joka2/NBR1 is also required for immunity against bacteria and oomycete pathogens, however, the extent to which defense-related autophagy acts against these pathogens is unknown (*Dagdas et al., 2018*; *Üstün et al., 2018*). Consistent with the important role of autophagy in plant immunity, adapted pathogens have evolved strategies to manipulate the host autophagy machinery (*Leary et al., 2018*; *Üstün et al., 2018*; *Leary et al., 2019*).

Plants detect pathogens via immune sensors consisting of surface localized pattern recognition receptors (PRRs) and intracellular nucleotide-binding leucine-rich repeat-containing proteins (NLRs). In response, pathogens secrete an arsenal of effector proteins to modulate host immunity and processes to support their virulence. Interestingly, effectors function not only to evade and suppress host immunity but also to mediate nutrient uptake (*Win et al., 2012*). Some filamentous plant pathogens, including the Irish potato famine pathogen *Phytophthora infestans*, produce hyphal extensions called haustoria that grow into the host cells to facilitate effector delivery and gain access to host nutrients (*Panstruga and Dodds, 2009*). A haustorium is a specialized infection structure that remains enveloped by an enigmatic host-derived membrane known as the extra-haustorial membrane (EHM), whose functions and biogenesis are poorly understood (*Whisson et al., 2016*). Notably, we previously showed that Joka2-mediated defense-related autophagy is diverted to the EHM during *P. infestans* infection (*Dagdas et al., 2018*). The pathogen counteracts this by deploying PexRD54, a host-translocated RXLR class of effector with five consecutive WY motifs, that targets plant autophagy (*Dagdas et al., 2016*). PexRD54 carries a canonical C-terminal ATG8 interacting motif (AIM) that is typically found on autophagy cargo receptors to bind ATG8 (*Maqbool et al., 2016*). Among the diverse set of potato ATG8 members (*Kellner et al., 2017*; *Zess et al., 2019*), PexRD54 preferentially binds the ATG8CL isoform and outcompetes Joka2/NBR1 from ATG8CL complexes, thereby disarming defense-related autophagy at the pathogen interface (*Dagdas et al., 2016*; *Dagdas et al., 2018*). Intriguingly, PexRD54 does not fully shutdown autophagy as has been shown for animal pathogens that suppress autophagy (*Choy et al., 2012*; *Kimmey and Stallings, 2016*; *Real et al., 2018*; *Xu et al., 2019*). Instead, it stimulates formation of autophagosomes that accumulate around the pathogen interface (*Dagdas et al., 2018*). How PexRD54 stimulates autophagy and in what way the pathogen benefits from this remains unknown.

Here, we show that PexRD54 mimics carbon starvation-induced autophagy by coupling the host vesicle transport regulator Rab8a (*Speth et al., 2009*; *Zheng et al., 2005*; *Pfeffer, 2017*) to autophagosome biogenesis at the pathogen interface. Unlike PexRD54 which activates autophagy by recruiting Rab8a to ATG8CL compartments, PexRD54's AIM peptide fails to associate with Rab8a, and instead functions as an autophagy inhibitor. Thus, by using an effector protein and its peptide derivative as molecular tools to perturb host-autophagy, we provide insights into not only how vesicle transport processes selectively support autophagosome formation, but also how the pathogen exploits these pathways to undermine plant immunity.

## Results

### PexRD54-ATG8 binding is not sufficient for stimulation of autophagosome formation

We have previously shown that PexRD54 stimulates formation of ATG8CL-autophagosomes in a macroautophagy dependent manner, and PexRD54 itself is a substrate of host autophagy when expressed in plant cells (*Dagdas et al., 2016*; *Maqbool et al., 2016*). During infection, these pathogen-induced autophagosomes are diverted to the EHM in a process that relies on the core autophagy machinery (*Dagdas et al., 2018*). Because AIM-mediated binding of PexRD54 to ATG8CL is essential for the activation of autophagosome formation by the effector protein (*Dagdas et al., 2016*; *Maqbool et al., 2016*), we reasoned that PexRD54 could stimulate autophagosome formation by either negative regulation of host autophagy suppressors or through positive regulation of host components to the autophagosome biogenesis sites.

To understand how PexRD54 stimulates autophagy, we used *N. benthamiana*, because: it is widely accepted as a solanaceous model plant in the field of plant-pathogen interactions; enables rapid functional, and cell biology assays using *Agrobacterium*; and can be infected by *P. infestans*. To first address whether ATG8 binding is sufficient to trigger autophagosome induction, we generated a PexRD54 truncate comprising only the C terminal AIM peptide (amino acids 350–381, hereafter AIMp), and compared its potency to stimulate autophagosome formation to the full-length protein. Strikingly, instead of stimulating autophagosome formation, AIMp fused to RFP (RFP:AIMp) significantly reduced the number of ATG8CL-autophagosomes in leaf epidermal cells (*Figure 1A–B*). Compared to RFP:GUS control, expression of RFP:AIMp reduced the number of GFP:ATG8CL-autophagosomes by ~sixfold, whereas cells expressing RFP:PexRD54 had a ~fourfold increase in GFP:ATG8CL-autophagosome numbers as has been shown before (*Figure 1A–B*; *Dagdas et al., 2016*). The AIMp interacted with ATG8CL in planta (*Figure 1C–D*), as was previously shown through in vitro studies (*Dagdas et al., 2016*; *Maqbool et al., 2016*). However, this association appears to take place mainly in the cytoplasm as the suppression of autophagosome formation by AIMp is such that we hardly observe any GFP:ATG8CL autophagosomes (*Figure 1A–B,E*). In contrast, RFP:PexRD54 and GFP:ATG8CL produced strong overlapping fluorescence signals which peak at mobile, ring-like ATG8CL-autophagosome clusters that are induced by PexRD54 as described previously (*Dagdas et al., 2016*). However, in cells expressing RFP:GUS control, we did not detect any RFP signal that peaks at ATG8CL-puncta (*Figure 1A–B,E*). Taken together these results show that binding of PexRD54 to ATG8CL, although necessary, is not sufficient to activate autophagosome biogenesis. This suggests while the full-length protein stimulates autophagy, PexRD54's AIM peptide functions as an autophagy suppressor.

## The AIM peptide of PexRD54 suppresses autophagy

To determine whether AIMp negatively regulates autophagy, we investigated its impact on autophagic flux by monitoring GFP:ATG8CL depletion over time. As there are no specific antibodies available for individual plant ATG8 isoforms, GFP:ATG8 depletion is used to measure autophagic flux in plants rather than quantifying the ratio of lipidated to unlipidated ATG8 isoforms as for other model organisms. Consistent with the AIMp triggered decrease in ATG8CL-autophagosome numbers (*Figure 1A–B*), RFP:AIMp stabilized GFP:ATG8CL compared to RFP:PexRD54 or the controls RFP:EV and RFP:GUS (*Figure 2A*, *Figure 2—figure supplement 1*). Western blotting showed that in the presence of RFP:AIMp, GFP:ATG8CL was still able to produce a strong protein signal even after 6 days of transient expression. On the other hand, GFP:ATG8CL protein signal was hardly detectable after just four days in the presence of RFP:PexRD54 or RFP:EV, indicating that the AIMp hampers autophagic flux (*Figure 2A*). Consistent with our previous report that PexRD54 acts as a cargo receptor that activates autophagy (*Dagdas et al., 2016*), we saw rapid decline of RFP:PexRD54 protein levels after 2 days post infiltration (dpi) with only free RFP detectable at six dpi, indicating vacuolar processing of RFP:PexRD54 (*Figure 2—figure supplement 1*). This was not due lack of PexRD54 expression at later time points as we could still detect mRNA expression of all constructs which are driven by constitutive 35S promoter (*Figure 2—figure supplement 2*). In line with this view, confocal microscopy revealed that while RFP:PexRD54 localizes to the cytoplasm at two dpi, RFP signal becomes vacuolar at 4 and 6 dpi (*Figure 2—figure supplement 3*), providing further evidence that PexRD54 itself is a substrate of autophagy and is degraded inside the plant vacuole, consistent with the flux assays we reported earlier (*Dagdas et al., 2016*; *Maqbool et al., 2016*). In contrast, RFP:AIMp remains mostly cytoplasmic at 2–6 dpi consistent with its function as a suppressor of ATG8 autophagy (*Figure 2—figure supplement 3*). However, we did not observe any stabilisation of the GFP control by RFP:AIMp, RFP:PexRD54 or RFP:GUS (*Figure 2—figure supplement 1*), indicating that reduced turnover of GFP:ATG8CL by RFP:AIMp is specific and is not due to altered Agrobacterium-mediated expression efficiency. Moreover, the AIM peptide mutated in the conserved ATG8-interacting region was unable to prevent ATG8CL depletion (*Figure 2—figure supplement 4*). Altogether, these data indicate that inhibition of autophagy by AIMp relies on ATG8 binding and AIMp can potentially block ATG8 dependent autophagy pathways.

We next investigated the extent to which AIMp acts on other potato ATG8 isoforms. RFP:AIMp showed a robust stabilization of all six potato ATG8 isoforms (*Figure 2B*), demonstrating that AIMp acts as a broad spectrum autophagy suppressor. To support these results, we measured the

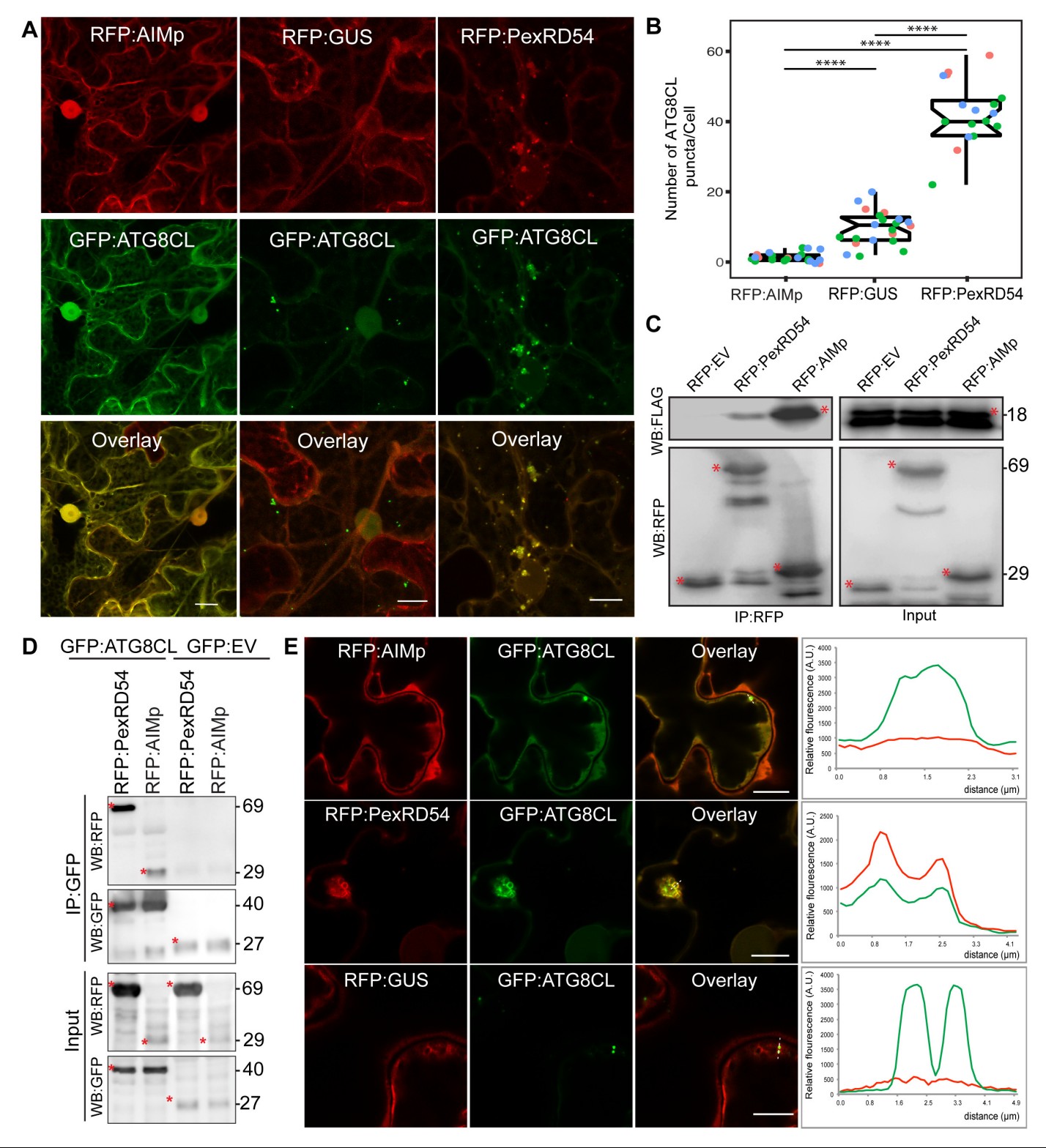

**Figure 1.** The AIM peptide of PexRD54 is not sufficient for stimulation of autophagosome formation. (**A**) Confocal micrographs of *Nicotiana benthamiana* leaf epidermal cells transiently expressing either RFP:AIMp (left), RFP:PexRD54 (middle), or RFP:GUS (right), with GFP:ATG8CL. Scale bars = 10 µm. Images shown are maximal projections of 26 frames with 1 µm steps (**B**) Quantification of autophagosome numbers from A shows RFP: PexRD54 expression increases ATG8CL autophagosomes per cell (40, *N* = 19 images quantified) compared to RFP:GUS control (10, *N = 22* images quantified), while RFP:AIMp significantly decreases ATG8CL autophagosome numbers (2, *N* = 23 images quantified). Scattered points show individual

*Figure 1 continued on next page*

*Figure 1 continued*

data points, color indicates biological repeats. Statistical differences were analyzed by Student t-test in R. Measurements were significant when p<0.05 (*) and highly significant when p<0.0001(****). (C) In planta co-immunoprecipitation between ATG8CL and PexRD54, AIMp or an Empty vector (EV). FLAG:ATG8CL was transiently co-expressed with either RFP:PexRD54, RFP:AIMp, or RFP:EV. IPs were obtained with anti-RFP antiserum. (D) Reverse pull-down between ATG8CL and PexRD54, AIMp, or GFP:EV. IPs were obtained with anti-GFP antiserum. (C– D) Red asterisks indicate expected band sizes. Protein sizes in kDa. (E) Confocal micrographs of RFP:AIMp (top), RFP:PexRD54 (middle), or RFP:GUS (bottom), co-expressed with GFP:ATG8CL. Transects in overlay panel correspond to plot of relative fluorescence over the labeled distance. RFP:PexRD54 co-localizes in discrete puncta with GFP: ATG8CL while RFP:AIMp and RFP:GUS fluorescence signals do not peak at GFP:ATG8CL puncta. Images shown are maximal projections of 12 frames with 1 μm steps. Scale bars represent 10 μm.

The online version of this article includes the following source data for figure 1:

**Source data 1.** Source data for western blots and autophagosome counts.

endogenous NBR1/Joka2 and ATG8 protein levels in *N. benthamiana* in the presence or absence of the AIMp. Consistently, ectopic expression of RFP:AIMp led to marked increase in both native NbJoka2 and NbATG8(s) levels compared to RFP:GUS expression, whereas RFP:PexRD54 mildly increased levels of native NbJoka2 but not NbATG8s (*Figure 2—figure supplement 5A*). These results further support the view that PexRD54's AIMp suppresses autophagy non-selectively, whereas PexRD54 activates ATG8CL-autophagy while neutralizing Joka2-mediated autophagy as shown before (*Dagdas et al., 2016*). Furthermore, AIMp enhanced protein levels of the recombinantly tagged derivatives of Joka2 and ATG8CL that are expressed under constitutive 35S promoter (*Figure 2—figure supplement 5B*), indicating that AIMp-triggered Joka2 and ATG8CL stabilization is not due to gene expression differences caused by the AIMp.

We then explored the potency of the AIMp in autophagy suppression when applied exogenously. For this we custom synthesized PexRD54's AIM peptide (AIMp$^{syn}$, 10 amino acids at the C terminus) along with an AIM peptide mutant (mAIMp$^{syn}$) that has a two amino acid substitution in the conserved residues of the AIM (*Dagdas et al., 2016*), fused to cell penetrating peptides. We first tested their activities in roots of a transgenic *N. benthamiana* line that stably express GFP:ATG8CL to test the uptake and efficacy of the peptide in different plant tissues and as many plant autophagy studies are performed in roots. Although both AIMp$^{syn}$ and mAIMp$^{syn}$ fused to 5-Carboxyfluorescein (CF-AIMp$^{syn}$ and CF-mAIMp$^{syn}$) were effectively taken up by the root cells (*Figure 2—figure supplement 6*), only the wild-type AIMp$^{syn}$ reduced the frequency of GFP:ATG8CL-puncta (by ~10-fold) compared to mAIMp$^{syn}$ or buffer control (*Figure 2C–D*). We also repeated these assays in leaf epidermal cells successfully, however, peptide translocation efficiency and thus autophagosome reduction by AIMp$^{syn}$ were much lower in leaves compared to root cells (*Figure 2E–F*, *Figure 2—figure supplement 7*). These findings demonstrate that PexRD54's AIM peptide suppresses autophagy in a variety of tissues, likely through binding to plant ATG8 isoforms with a high affinity and limiting their access to the autophagy adaptors that are essential for induction of autophagy. This implies that full-length PexRD54 carries additional features to stimulate autophagosome formation, by for instance, recruiting and/or manipulating other host components.

## PexRD54 associates with the host vesicle transport regulator Rab8a independent of ATG8CL binding

We next set out to investigate the mechanism of autophagy activation by PexRD54. Although the underlying molecular mechanisms are largely unknown, autophagosome biogenesis relies on vesicle trafficking and fusion events in yeast and animals (*Singh et al., 2019*; *Nair et al., 2011*). We therefore reasoned that in addition to binding ATG8CL, PexRD54 could possibly hijack host vesicle transport machinery to stimulate autophagosome biogenesis. Interestingly, our previous proteomics survey identified Rab8a, which shows the most sequence similarity to Arabidopsis RabE1a, a member of the small Ras-related GTPases that mediate vesicle transport and fusion events, as a candidate PexRD54 interactor (*Dagdas et al., 2016*). We first validated PexRD54-Rab8a association through co-immunoprecipitation assays by co-expressing the potato Rab8a (herein Rab8a) or the *N. benthamiana* Rab8a (NbRab8a) with PexRD54 in planta (*Figure 3A*, *Figure 3—figure supplement 1*). Notably, the AIM mutant of PexRD54 (PexRD54$^{AIM}$) that cannot bind ATG8CL still interacted with Rab8a to a similar degree as PexRD54 (*Figure 3A*), indicating that PexRD54 associates with Rab8a independent of its ATG8CL-binding activity. Consistent with this, the AIMp failed to associate

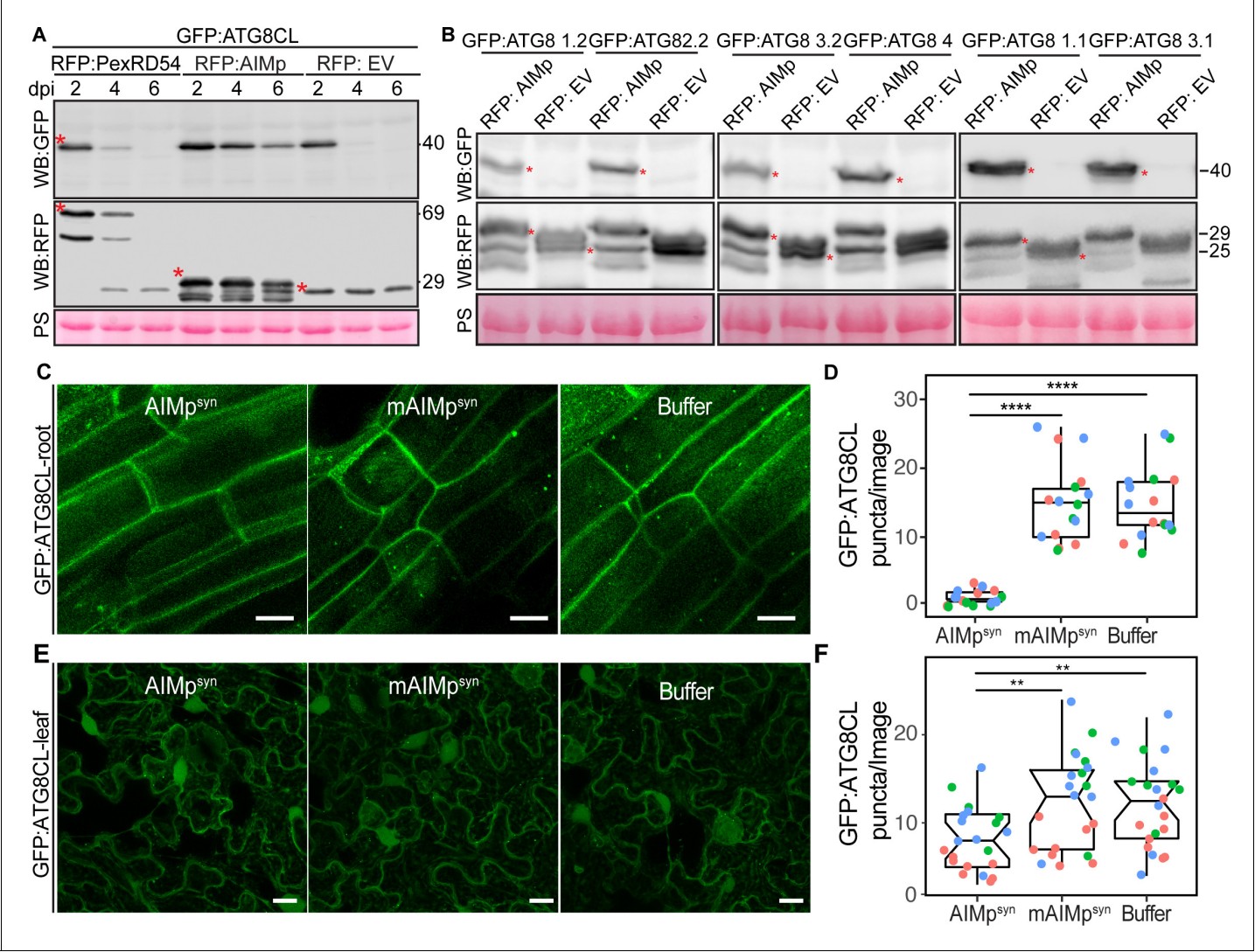

**Figure 2.** The AIM peptide supresses autophagy flux. (**A**) Western blots show depletion of GFP:ATG8CL is substantially reduced by RFP:AIMp compared to RFP:PexRD54 and RFP:GUS control beyond 2 days post infiltration. (**B**) Western blots show various Agrobacterium expressed GFP:ATG8 isoforms is stabilized by RFP:AIMp. Total protein extracts were prepared 4 days post Agroinfiltration. Red asterisks in A-B show expected band sizes. Protein sizes in kDa. (**C**) Confocal micrographs of transgenic *N. benthamiana* root cells stably expressing GFP-ATG8CL infiltrated with cell penetrating peptides or buffer. Images shown are maximal projections of 10 frames with 1 μm steps. (**D**) Compared to a buffer control (15, *N* = 16 images quantified), synthesized AIM peptide fused to a cell penetrating peptide (AIMpsyn) suppresses ATG8CL autophagosomes per image in roots (1, *N* = 18 images quantified), while the AIM peptide mutant mAIMp$^{synAA}$ does not (15, *N* = 17 images quantified). Statistical differences were analyzed by Student t-test in R. Measurements were significant when p<0.05 (*) and highly significant when p<0.0001(****). (**E**) Transgenic *N. benthamiana* leaf epidermal cells stably expressing GFP-ATG8CL infiltrated with cell penetrating peptides or buffer. Images shown are maximal projections of 25 frames with 1 μm steps. Scale bars represent 10 μm. (**F**) Scatter-boxplot shows exogenous application of cell penetrating AIMp$^{syn}$ in GFP-ATG8CL transgenic *N. benthamiana* significantly decreases the number of ATG8 puncta per image (8, *N* = 21 images quantified) compared to cell penetrating mAIM$^{synAA}$ (12, *N* = 22 images quantified), or Buffer control (12, *N* = 22 images quantified). Scattered points show individual data points, color indicates biological repeat. Statistical differences were analyzed by Students t-test in R. Measurements were significant when p<0.05 (*) and highly significant when p<0.0001(****).

The online version of this article includes the following source data and figure supplement(s) for figure 2:

**Source data 1.** Source data for western blots and autophagosome counts.
**Figure supplement 1.** AIMp stabilizes ATG8CL but not GFP.
**Figure supplement 2.** RT-PCR shows expression of RFP constructs.
**Figure supplement 3.** PexRD54 is trafficked to the vacuole while AIMp remains cytoplasmic.
**Figure supplement 4.** AIM peptide mutant does not stabilize ATG8CL.
**Figure supplement 5.** AIMp stabilizes both endogenous and transiently expressed NBR1/Joka2 and ATG8(s).
*Figure 2 continued on next page*

*Figure 2 continued*

**Figure supplement 6.** Cell penetrating AIM peptide constructs are uptaken in roots.
**Figure supplement 7.** Cell penetrating AIM peptides translocate inside leaf cells.

with Rab8a in pull down assays, although it still strongly interacted with ATG8CL (*Figure 3B*). These results suggest that PexRD54's N-terminal region preceding the C-terminal AIM mediates Rab8a association.

We then investigated the subcellular distribution of PexRD54 and Rab8a through confocal microscopy in leaf epidermal cells. Both stably and transiently expressed Rab8a fused to GFP (GFP:Rab8a) produced fluorescent signals at both the plasma membrane and the vacuolar membrane (tonoplast) (*Figure 3—figure supplement 2*). In addition, GFP:Rab8a localized to mobile puncta (0.2–0.5 µm in diameter) as well as to larger ring-shaped structures (*Figure 3—figure supplement 2*, *Video 1*), indicating that Rab8a could be involved in multiple cellular trafficking events. To determine the subcellular compartment(s) where PexRD54 associates with Rab8a, we performed co-localisation experiments of GFP:Rab8a with RFP:PexRD54, RFP:AIMp or RFP:GUS. In line with the pull-down assays (*Figure 3A–B*), a subset of punctate structures labeled by GFP:Rab8a showed a clear overlap with RFP:PexRD54-puncta, whereas we did not detect any RFP signal peaking at GFP:Rab8a puncta in cells expressing RFP:AIMp or RFP:GUS (*Figure 3C*). Together, these results demonstrate that PexRD54 associates with Rab8a in an AIM independent manner and raise the possibility that Rab8a could be an important component of PexRD54 driven autophagy.

## PexRD54 shows a higher affinity towards Rab8a-S29N mutant

Because Rab GTPases function by converting between GTP and GDP bound states, we decided to generate Rab mutants that mimic the active (GTP) and inactive (GDP bound) conformations, which are helpful for characterization of the Rab GTPase functions. Although earlier work challenged the applicability of these mutations (*Langemeyer et al., 2014*; *Nottingham and Pfeffer, 2014*), we reasoned that Rab8a mutants could still be useful to dissect the role of Rab8a in PexRD54 activated autophagy. To determine whether PexRD54 favors a particular form of Rab8a, we produced Rab8a point mutants that we presume to mimic the GTP (Rab8a$^{Q74L}$) or GDP (Rab8a$^{S29N}$) bound states and investigated their subcellular distribution (*Figure 3—figure supplement 3*). Unlike GFP:Rab8a, which predominantly labeled the plasma membrane, GFP:Rab8a$^{S29N}$ mutant showed an even distribution at the plasma membrane and the tonoplast (*Figure 3—figure supplement 2B–C*). In addition, both GFP:Rab8a$^{S29N}$ and GFP:Rab8a marked punctate structures with varying size and shape (*Figure 3—figure supplement 3B–C*). In contrast, GFP:Rab8a$^{Q74L}$ was mainly trapped in the tonoplast and showed reduced punctate distribution compared to GFP:Rab8a or GFP:Rab8a$^{S29N}$ (*Figure 3—figure supplement 3B–D*), indicating that the Q74L mutant may not be representing the fully active form of Rab8a as previously reported for other Rab GTPases (*Langemeyer et al., 2014*; *Nottingham and Pfeffer, 2014*).

We next examined the extent to which Rab8a mutants colocalize with PexRD54. When co-expressed with BFP:PexRD54, both GFP:Rab8a and GFP:Rab8a$^{S29N}$ consistently produced sharp fluorescence signals that overlap with the typical ring-like autophagosomes marked by PexRD54 (*Figure 3—figure supplement 4A–B*). However, GFP:Rab8a$^{Q74L}$ showed a similar localization pattern to the GFP control, and mostly did not produce fluorescence signals that peak at BFP:PexRD54-puncta (*Figure 3—figure supplement 4C–D*). We quantified these observations in multiple independent experiments where GFP:Rab8a and GFP:Rab8a$^{S29N}$ frequently (68%, N = 23) labeled BFP:PexRD54-puncta, whereas GFP:Rab8a$^{Q74L}$ only did so significantly less often (25%, N = 20) (*Figure 3—figure supplement 4E*). As an additional control, we also checked for colocalization between Rab8a mutants and PexRD54's AIM peptide. However, we did not observe any puncta co-labeled by RFP:AIMp and GFP:Rab8a or any of the Rab8a mutants we tested (*Figure 3—figure supplement 5*). These observations are consistent with the results that PexRD54's AIM peptide fails to associate with Rab8a (*Figure 3B*) and suppresses autophagosome formation (*Figures 1–2*). We then compared the binding affinity of PexRD54 to Rab8a and its mutants via in planta co-immunoprecipitation. Rab8a$^{S29N}$ pulled-down PexRD54 more than wild type GFP:Rab8a or GFP:Rab8a$^{Q74L}$ in planta

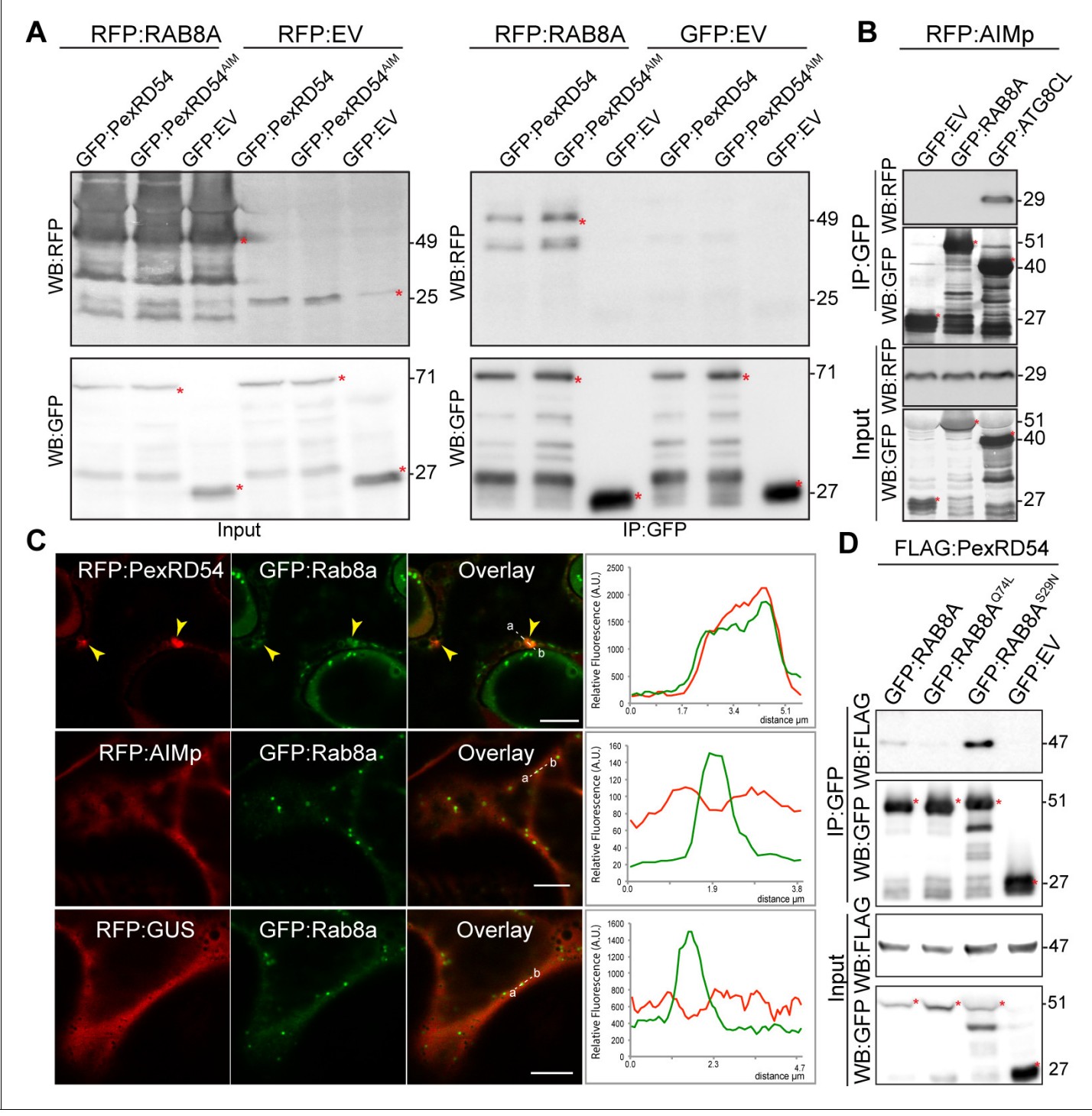

**Figure 3.** PexRD54 associates with the host vesicle transport regulator Rab8a independently of its ATG8CL binding. (**A**) In planta co-immunoprecipitation between Rab8a and PexRD54 or PexRD54[AIM]. RFP:Rab8a was transiently co-expressed with either GFP:EV, GFP:PexRD54 or GFP: PexRD54[AIM]. Red asterisks indicate expected band sizes. (**B**) In planta co-immunoprecipitation between the AIMp and ATG8CL or Rab8a. RFP:AIMp was transiently co-expressed with either GFP:EV, GFP:ATG8CL, or GFP:Rab8a. IPs were obtained with anti-GFP antiserum. Total protein extracts were immunoblotted. Red asterisks indicate expected band sizes. (**C**) Confocal micrographs of *N. benthamiana* leaf epidermal cells transiently expressing either RFP:PexRD54 (top), RFP:AIMp (middle), or RFP:GUS (bottom), with GFP:Rab8a. Yellow arrows show colocalization between constructs. Transects in overlay panel correspond to plot of relative fluorescence over the labeled distance. RFP:PexRD54 co-localizes in discrete punctate structures with GFP:Rab8a while RFP:AIMp and RFP:GUS show diffuse expression. Scale bars, 5 µm (**D**).

The online version of this article includes the following source data and figure supplement(s) for figure 3:

*Figure 3 continued on next page*

*Figure 3 continued*

**Source data 1.** Source data for western blots, Rab8a/mutant sequences, autophagosome counts, and localization data sheets.
**Figure supplement 1.** PexRD54 interacts with NbRab8a.
**Figure supplement 2.** Rab8a localizes to the tonoplast, plasma membrane, and punctate structures.
**Figure supplement 3.** Subcellular distribution of Rab8a and its mutants.
**Figure supplement 4.** PexRD54 puncta preferentially co-localize with puncta of wild type and GDP bound form of Rab8a (Rab8a$^{S29N}$) rather than GTP bound form (Rab8a$^{Q74L}$).
**Figure supplement 5.** Rab8a$^{S29N}$ and GFP:Rab8a$^{Q74L}$ do not co-localize with AIM peptide in planta.

(*Figure 3D*), suggesting that PexRD54 preferentially associates with the GDP bound state of Rab8a (S29N).

### In planta

co-immunoprecipitation of PexRD54 with Rab8a, Rab8a$^{Q74L}$, Rab8a$^{S29N}$, or GFP. FLAG:PexRD54 was transiently co-expressed with GFP:Rab8a, GFP:Rab8a$^{Q74L}$, GFP:Rab8a$^{S29N}$, or GFP:EV. IPs were obtained with anti-GFP antiserum and total protein extracts were immunoblotted with GFP and FLAG antisera. Red asterisks indicate expected band sizes.

### PexRD54 recruits Rab8a to autophagosome biogenesis sites

We next explored the potential role of Rab8a in PexRD54-triggered autophagy. To this end, we first investigated the extent to which PexRD54 associates with its two host interactors, Rab8a and ATG8CL. We achieved this through live-cell imaging of Rab8a and ATG8CL co-expressed in combination with either PexRD54, AIMp, or BFP control. This revealed that BFP:PexRD54, but not free BFP or BFP:AIMp, localizes to puncta co-labeled by RFP:ATG8CL and GFP:Rab8a (*Figure 4A–C*). Furthermore, GFP:Rab8a localized to ring-shaped RFP:ATG8CL clusters triggered by BFP:PexRD54, whereas no such structures occurred in cells expressing BFP control or BFP:AIMp (*Figure 4A–C*). Notably, our quantitative imaging revealed that, even in the absence of PexRD54, more than half of RFP:ATG8CL-puncta (60%, *N* = 18 images) are positively labeled by GFP:Rab8a (*Figure 4A,D*). However, BFP:PexRD54 expression significantly increased the frequency of GFP:Rab8a-positive RFP:ATG8CL-puncta (85%, *N* = 18 images) (*Figure 4B,D*). Conversely, we rarely detected any fluorescent puncta that were co-labeled by GFP:Rab8a and RFP:ATG8CL in the presence of BFP:AIMp (6%, *N* = 18), which strongly suppresses autophagosome formation (*Figure 4C–D*). These results indicate that a subset of Rab8a localizes to autophagy compartments marked by ATG8CL and this is enhanced by PexRD54. Consistently, in plants stably expressing GFP:Rab8a, we observed a similar degree of PexRD54-triggered increase in ATG8CL-Rab8a colocalization in ring-shaped ATG8CL-clusters (*Figure 4—figure supplement 1*), suggesting that PexRD54 might boost Rab8a recruitment to autophagic compartments. To then gain biochemical evidence for PexRD54-mediated recruitment of Rab8a to ATG8CL compartments, we conducted in planta co-immunoprecipitation assays between Rab8a and ATG8CL in presence of PexRD54, AIMp, or a control. We extracted proteins at an early time point (2 dpi) to minimize differences in ATG8CL levels, especially because we express proteins that alter autophagic

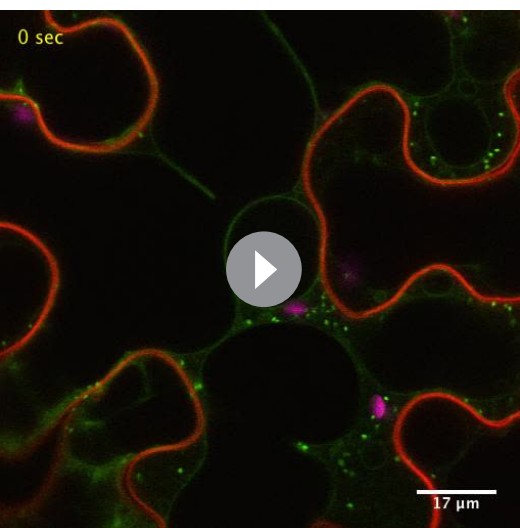

**Video 1.** Rab8a localizes to small mobile vesicles and large ring-shaped structures. GFP:Rab8a is co-expressed with the EHM marker RFP:REM1.3 via agroinfiltration in *N. benthamiana* leaf epidermal cells. Confocal laser scanning microscopy was used to monitor Rab8a-labeled vesicles 3 days post infiltration. The movie represents time-lapse of 76 frames acquired during 3 min 48 s (Frame interval: 3 s).
https://elifesciences.org/articles/65285#video1

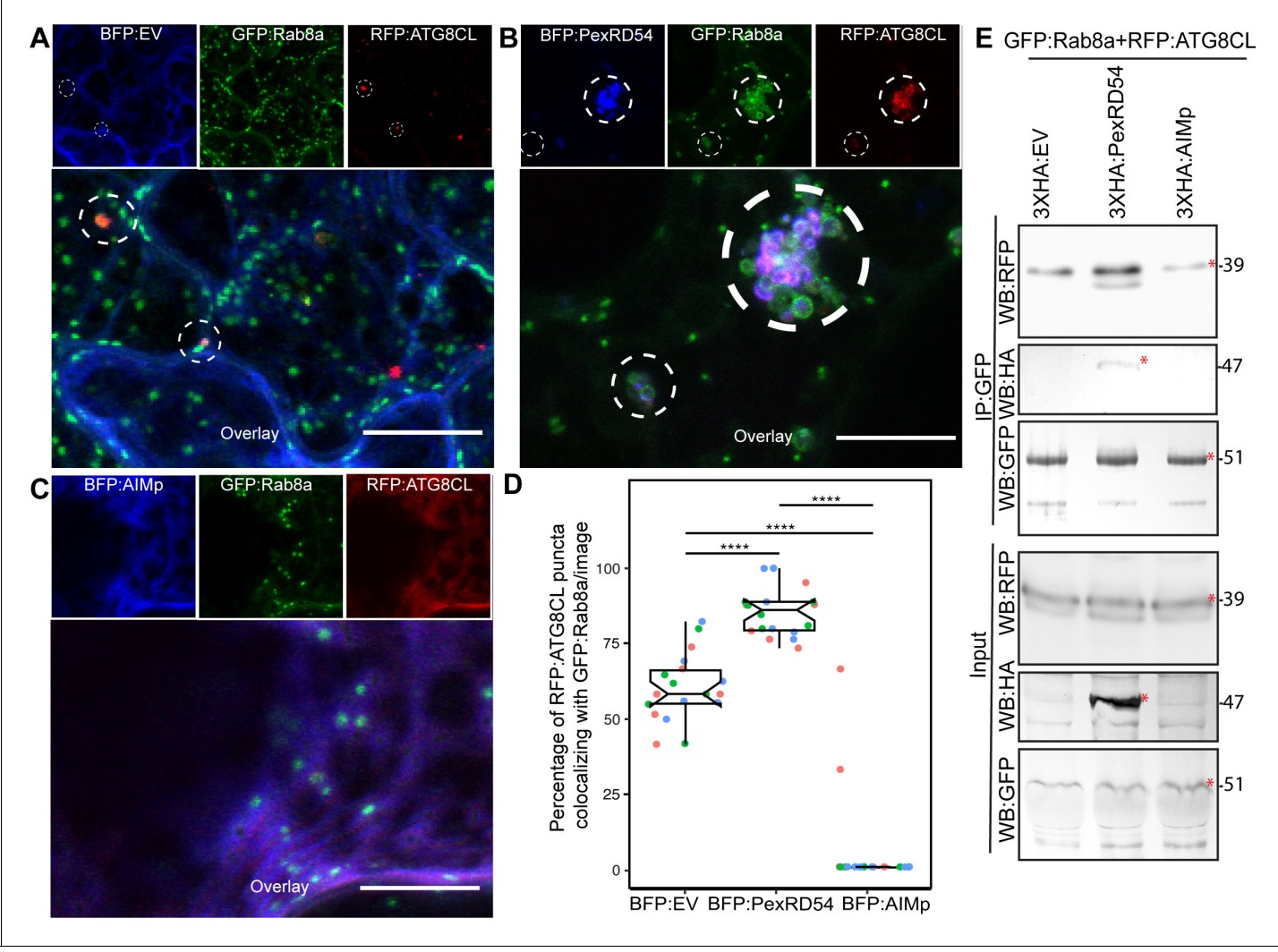

**Figure 4.** PexRD54 recruits Rab8A to ATG8CL-labeled autophagosomes. (**A–C**) Maximum projection confocal micrographs of *N. benthamiana* leaf epidermal cells transiently expressing either BFP:EV (**A**), BFP:PexRD54 (**B**), or BFP:AIMp (**C**), with GFP:Rab8a and RFP:ATG8CL. Dashed white circles show variable colocalization between RFP:ATG8CL and GFP:Rab8a. Scale bars represent 10 μm (**D**) BFP:PexRD54 expression significantly increases punctate colocalization between RFP:ATG8CL and GFP:Rab8 (85%, $N$ = 18 images quantified), while BFP:AIMp significantly reduces colocalization between RFP:ATG8CL and GFP:Rab8 (6%, $N$ = 18 images quantified) compared to the BFP:EV control (60%, $N$ = 18 images quantified). Scattered points show individual data points, colors indicate biological repeats. Statistical differences were analyzed by Welch Two Sample t-test in R. Measurements were significant when $p<0.05$ (*) and highly significant when $p<0.0001$(****). (**E**) In planta co-immunoprecipitation between Rab8A and ATG8CL, and PexRD54 or AIMp. GFP:Rab8A and RFP:ATG8CL were transiently co-expressed with either 3xHA:EV, 3xHA:PexRD54, or 3xHA:AIMp. IPs were obtained with anti-GFP antiserum. Total protein extracts were immunoblotted. Red asterisks indicate expected band sizes. Protein sizes in kDa. The online version of this article includes the following source data and figure supplement(s) for figure 4:

**Source data 1.** Source data for western blots, colocalization data sheets and image analysis plugin.
**Figure supplement 1.** PexRD54 increases Rab8a and ATG8CL co-localisation.
**Figure supplement 2.** RT-PCR shows expression of 3XHA constructs.
**Figure supplement 3.** PexRD54 enhances proximity of Rab8a puncta to ATG9 compartments.

activity (PexRD54 and AIMp). We observed noticeably more RFP:ATG8CL pulled down with GFP: Rab8a in presence of 3xHA:PexRD54 but not 3xHA:AIMp or 3xHA:EV (*Figure 4E*). As the 3xHA tag is only 27aa and 3.3 kDa, expression of smaller constructs such as 3xHA:EV and 3xHA:AIMp are not visible on western blots. Therefore, we validated expression of the HA-tagged constructs by RT-PCR using the RNA extracts from the Agroinfiltrated leaf patches (*Figure 4—figure supplement 2*).

Altogether, these results show that PexRD54 enhances Rab8a accumulation at ATG8CL-autophagosomes.

To further ascertain the functional relationship between PexRD54 and Rab8a, we investigated the degree to which Rab8a associates with autophagy machinery. We monitored the association of RFP:Rab8a with the early autophagosome biogenesis marker protein ATG9:GFP in combination with BFP:PexRD54, BFP:AIMp, or BFP. Confocal microscopy analyses revealed that ATG9:GFP puncta frequently associate with RFP:Rab8a-labeled vesicles. However, we detected an increased incidence of RFP:Rab8a puncta that are in contact with the mobile ATG9:GFP compartments (*Video 2*) in the presence of BFP:PexRD54 (68%, *N* = 44) compared to free BFP (38%, *N* = 31) or BFP:AIMp (31%, *N* = 58) (*Figure 4—figure supplement 3*), indicating that PexRD54 stimulates association of Rab8a with the autophagosome biogenesis machinery. Notably, in the presence of BFP:PexRD54, but not BFP:AIMp or BFP:EV, ATG9:GFP puncta showed more proximity to RFP:Rab8a puncta (*Figure 4—figure supplement 3*). Furthermore, time-lapse microscopy revealed that these mobile ATG9:GFP compartments co-migrate with BFP:PexRD54/RFP:Rab8a-positive puncta (*Video 2*). These results implicate Rab8a in autophagy and indicate that PexRD54 promotes Rab8a recruitment to autophagosome biogenesis sites.

## Rab8a is required for PexRD54-triggered autophagy

We next investigated whether Rab8a is required for PexRD54-mediated autophagy. We measured the impact of *NbRab8a* silencing on autophagy by quantifying the RFP:ATG8CL-autophagosome numbers. In the *N. benthamiana* genome, we identified at least four genes encoding full-length Rab8a like proteins (*NbRab8a1-4*). We first decided to generate a RNA interference (RNAi) construct (RNAi:NbRab8a[1-2] hereafter) that can target the three prime untranslated regions (UTR) of *NbRab8a1* and *NbRab8a2*, the two closest homologs of the potato *Rab8a* found in *N. benthamiana*. RNAi:NbRab8a[1-2] showed specific silencing of the *NbRab8a1-2* but not *NbRab8a3 and NbRab8a4* (*Figure 5—figure supplement 1*). In the absence of PexRD54, silencing of *NbRab8a1-2* did not alter the number of RFP:ATG8CL puncta (*Figure 5—figure supplement 2*). However, following stimula-

tion of autophagy by transient expression of GFP:PexRD54, the number of RFP:ATG8CL-puncta/cell in RNAi:NbRab8a[1-2] background reduced by half compared to a RNAi:GUS control (*Figure 5—figure supplement 2*). This suggests that simultaneous knockdown of *NbRab8a1* and *NbRab8a2* does not affect basal autophagy, but negatively impacts PexRD54-triggered autophagy. To validate these results, we set up a complementation assay in which we silenced *NbRab8a1-2* in transgenic *N. benthamiana* lines stably expressing the GFP tagged potato Rab8a, which evades RNA silencing because it lacks the three prime UTR targeted by the RNAi:NbRab8a[1-2] construct. Consistent with the results obtained in *Figure 5—figure supplement 2*, we detected greater than two-fold decrease in the number of HA:PexRD54 triggered RFP:ATG8CL-puncta upon delivery of RNAi:NbRab8a[1-2] construct in wild-type plants compared to RNAi:GUS (*Figure 5A–B*, *Figure 5—figure supplement 3*). On the other hand, the frequency of RFP:ATG8CL-puncta is not altered by RNAi:NbRab8a[1-2] in cells expressing the HA vector control (*Figure 5A–B*, *Figure 5—figure supplement 3*). We then set up a genetic complementation assay using stable transgenic *N. benthamiana* lines expressing the potato GFP:Rab8a protein that is resistant to

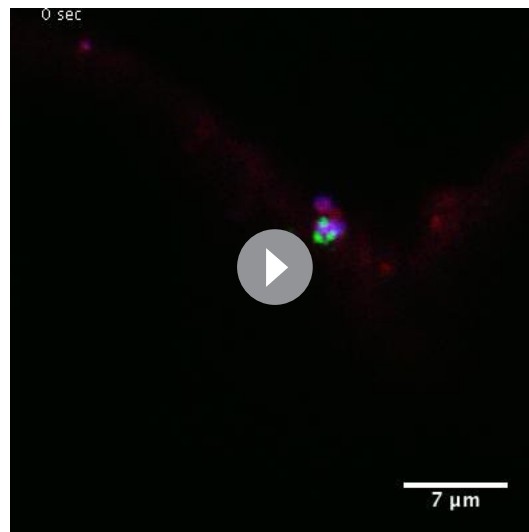

**Video 2.** Mobile PexRD54/Rab8a-positive puncta co-migrate with ATG9 vesicles. RFP:Rab8a, BFP:PexRD54, and ATG9-GFP are co-expressed three via agroinfiltration in *N. benthamiana* leaf epidermal cells. Confocal laser scanning microscopy was used to monitor PexRD54/Rab8a-labeled vesicles and ATG9 compartments 3 days post infiltration. The movie represents time-lapse of 28 frames acquired during 4 min 12 s (Frame interval: 9 s).
https://elifesciences.org/articles/65285#video2

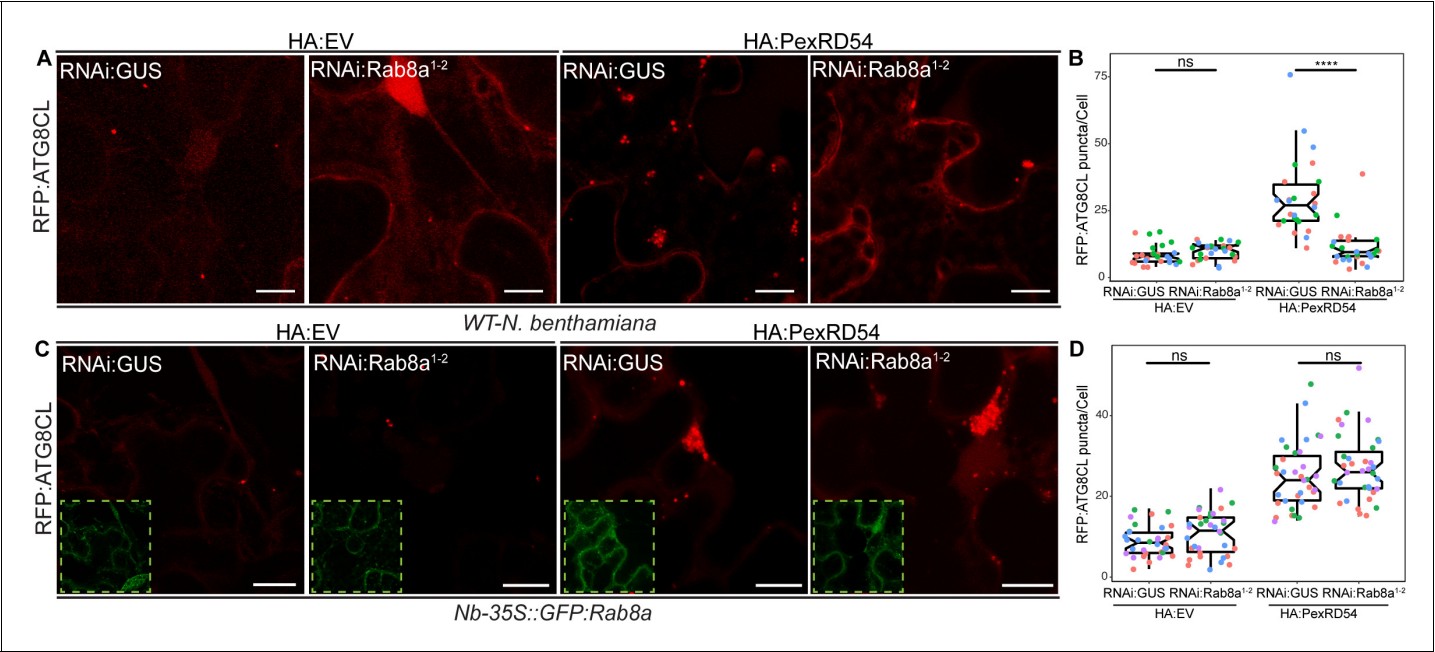

**Figure 5.** Rab8a is required for PexRD54-mediated autophagosome formation. (**A–B**) RNAi-mediated silencing of Rab8a leads to reduction of autophagosome numbers induced by PexRD54. (**A**) Confocal micrographs of *N. benthamiana* leaf epidermal cells transiently expressing RFP:ATG8CL with HA:EV or HA:PexRD54 either combined with RNAi:GUS or RNAi:NbRab8a$^{1-2}$. Images shown are maximal projections of 25 frames with 1.2 μm steps. Scale bars represent 10 μm. (**B**) Silencing *N. benthamiana Rab8a1-2* significantly suppresses the autophagosome formation induced by PexRD54 (11, N = 26 images quantified) compared to GUS silencing control (30, N = 26 images quantified), but in the absence of PexRD54 silencing *Rab8a1-2* has no effect on endogenous autophagosome number (10, N = 27 images quantified) compared to silencing control (9, N = 24 images quantified). Statistical differences were analyzed by Welch Two Sample t-test in R. Measurements were significant when p<0.05 (*) and highly significant when p<0.0001(****). (**C-D**) Complementation of *Rab8a1-2* silencing phenotype. (**C**) Confocal micrographs of GFP:NbRab8a leaf epidermal cells transiently expressing RFP:ATG8CL with HA:EV or HA:PexRD54 either combined with RNAi:GUS or RNAi:NbRab8a$^{1-2}$. Dashed white squares show GFP signal of complemented *GFP:Rab8a*. Images shown are maximal projections of 25 frames with 1.2 μm steps. Scale bars represent 10 μm. (**D**) Complementing endogenous *Rab8a1-2* silencing in the silencing resistant *Nb-35S::GFP:Rab8a* transgenics recovered PexRD54-induced autophagosome formation (24, N = 35 images quantified), to similar levels to the silencing control RNAi:GUS (28, N = 35 images quantified). Statistical differences were analyzed by Welch Two Sample t-test in R. Measurements were significant when p<0.05 (*) and highly significant when p<0.0001(****).

The online version of this article includes the following source data and figure supplement(s) for figure 5:

**Source data 1.** Source data for Rab8a sequences and autophagosome counts.
**Figure supplement 1.** Validation of *NbRab8a-1/2* silencing by RNAi:NbRab8a$^{1-2}$.
**Figure supplement 2.** PexRD54 increases the number of ATG8CL puncta in a Rab8a-dependent manner.
**Figure supplement 3.** Validation of silencing of *NbRab8a-1* by RNAi:NbRab8a$^{1-2}$.
**Figure supplement 4.** Validation of RNAi:NbRab8a$^{1-4}$ silencing construct.
**Figure supplement 5.** Silencing of *NbRab8a1-4* significantly reduces ATG8CL autophagosome numbers.
**Figure supplement 6.** Silencing of *NbRab8a1-4* significantly reduces PexRD54-triggered ATG8CL autophagosome numbers.
**Figure supplement 7.** Dominant negative mutant of Rab8a (N128I) decreases PexRD54-induced autophagosome formation.
**Figure supplement 8.** Rab8a (N128I) reduces the number of PexRD54-induced autophagosomes.
**Figure supplement 9.** WT Rab8a and Rab8a$^{S29N}$ increases the amount of PexRD54-triggered puncta.

silencing by RNAi:NbRab8a$^{1-2}$ construct. In stable transgenic plants expressing the silencing resistant potato GFP:Rab8a protein, RNAi:NbRab8a$^{1-2}$ did not change the number of RFP:ATG8CL puncta with or without HA:PexRD54, compared to cells that express RNAi:GUS control (*Figure 5C–D*, *Figure 5—figure supplement 3*). These results suggest that Rab8a positively regulates PexRD54-mediated autophagy.

We next generated a hairpin-silencing construct (RNAi:NbRab8a1-4) that targets all four *Rab8a* members in *N. benthamiana*. The RNAi:NbRab8a1-4 construct substantially silenced *NbRab8a1* and *NbRab8a3* while silencing *NbRab8a2* and *NbRab8a4* to a lesser extent (*Figure 5—figure supplement 4*). RNAi:NbRab8a1-4 did not however silence an unrelated Rab GTPase family member Rab11 (*Figure 5—figure supplement 4*). Intriguingly, knockdown of the four Rab8a isoforms using the

RNAi:NbRab8a[1-4] construct significantly reduced basal ATG8CL (*Figure 5—figure supplement 5*) and PexRD54-triggered ATG8CL autophagosome numbers (*Figure 5—figure supplement 6*). This suggests a potential redundancy in Rab8a function in basal autophagy. Alternatively, *NbRab8a3-4* might be involved in basal autophagy, whereas *NbRab8a1-2* are not, which needs to be tested in future.

To gain further genetic evidence for Rab8a's positive role in PexRD54-triggered autophagosome formation, we used the dominant negative Rab8a mutant (N128I) (*Essid et al., 2012*) and measured its impact on formation of RFP:ATG8CL-autophagosomes in the presence or absence of HA:PexRD54. Consistent with the silencing assays (*Figure 5A–D*), GFP:Rab8a[N128I] led to 50% reduction in PexRD54 triggered ATG8CL-autophagosome numbers compared to wild-type GFP:Rab8a (*Figure 5—figure supplement 7*). We further validated the dominant negative role of GFP:Rab8a[N128I], which significantly reduced PexRD54-triggered ATG8CL puncta compared to GFP control in two independent biological replicates (*Figure 5—figure supplement 8*).

Since we found that PexRD54 associates with Rab8a and its mutants with varying affinities (*Figure 3D*), we next checked whether ectopic expression of Rab8a and its mutants (S29N and Q74L) have any effect on the formation of PexRD54-autophagosomes. Compared to GFP control, GFP:Rab8a expression led to a slight increase (~1.5 fold) in the number of BFP:PexRD54 puncta (*Figure 5—figure supplement 9*), suggesting that Rab8a could positively regulate autophagosome formation. Expression of GDP-bound GFP:Rab8a[S29N] substantially enhanced (~three fold) the frequency of BFP:PexRD54 puncta compared to a GFP control, whereas GTP bound GFP:Rab8a[Q74L] did not lead to any significant changes in the number of BFP:PexRD54 puncta compared to the GFP control (*Figure 5—figure supplement 9*). These results are consistent with the pulldown assays, which revealed stronger interaction between PexRD54 and Rab8a[S29N]. These findings were initially surprising given the general view that Rab S-to-N mutations in this position lead to less active (or inactive) forms that mimic the GDP-bound state, whereas the Q-to-L mutations in this site are assumed to be locked in GTP-bound state that is more active. However, there are reports which revealed that these mutations cannot be generalized (*Langemeyer et al., 2014*; *Nottingham and Pfeffer, 2014*). Consistent with this view, the GDP-bound form of the mammalian Rab8a (T22N mutant) was found to promote lipid droplet (LD) fusions, indicating that S-to-N mutation in this Rab protein functions differently. Therefore, further biochemical evidence is required to determine whether these mutations show perturbed GTPase activities. Nevertheless, together with the data presented in *Figure 3D*, these findings demonstrate that PexRD54-driven autophagy requires Rab8a.

## Rab8a is specifically recruited to PexRD54-autophagosomes and is dispensable for Joka2-mediated autophagy

To better characterize the autophagy pathway stimulated by PexRD54, we further investigated the interplay between Rab8a and ATG8CL. The weak interaction of ATG8CL and Rab8a in the absence of PexRD54 (*Figure 4E*) suggests for an indirect association potentially mediated through a host autophagy adaptor. Therefore, we explored whether increased ATG8CL and Rab8a association triggered by PexRD54 is a general hallmark of autophagy activation or is a process that is stimulated through plant selective autophagy adaptors. Because the plant autophagy cargo receptor Joka2 that mediates aggrephagy also binds ATG8CL and stimulates autophagosome formation (*Dagdas et al., 2016*; *Jung et al., 2020*), we tested if Joka2 could interact with Rab8a and enhance Rab8a-ATG8CL association. Unlike PexRD54, Joka2 did not colocalize or interact with Rab8a (*Figure 6A–B*). Moreover, our quantitative imaging revealed that Joka2 overexpression leads to a reduction of RFP:ATG8CL puncta positively labeled by GFP:Rab8a (*Figure 6C,D*). This sharply contrasts with the positive impact of PexRD54 on ATG8CL-Rab8a association (*Figure 6C,D*), indicating that the autophagy pathway mediated by Joka2 is different from the PexRD54 triggered autophagy, and possibly does not require Rab8a function. Supporting this, we did not detect any difference in formation of Joka2-triggered autophagosomes upon *NbRab8a1-2* silencing compared to *GUS* silencing (*Figure 6E–F*, *Figure 6—figure supplement 1*). Collectively, these results indicate that Joka2-mediated autophagy pathway does not involve Rab8a, and the weak association between ATG8CL and Rab8a observed in the absence of PexRD54 is not mediated by Joka2 but potentially through an unknown autophagy adaptor.

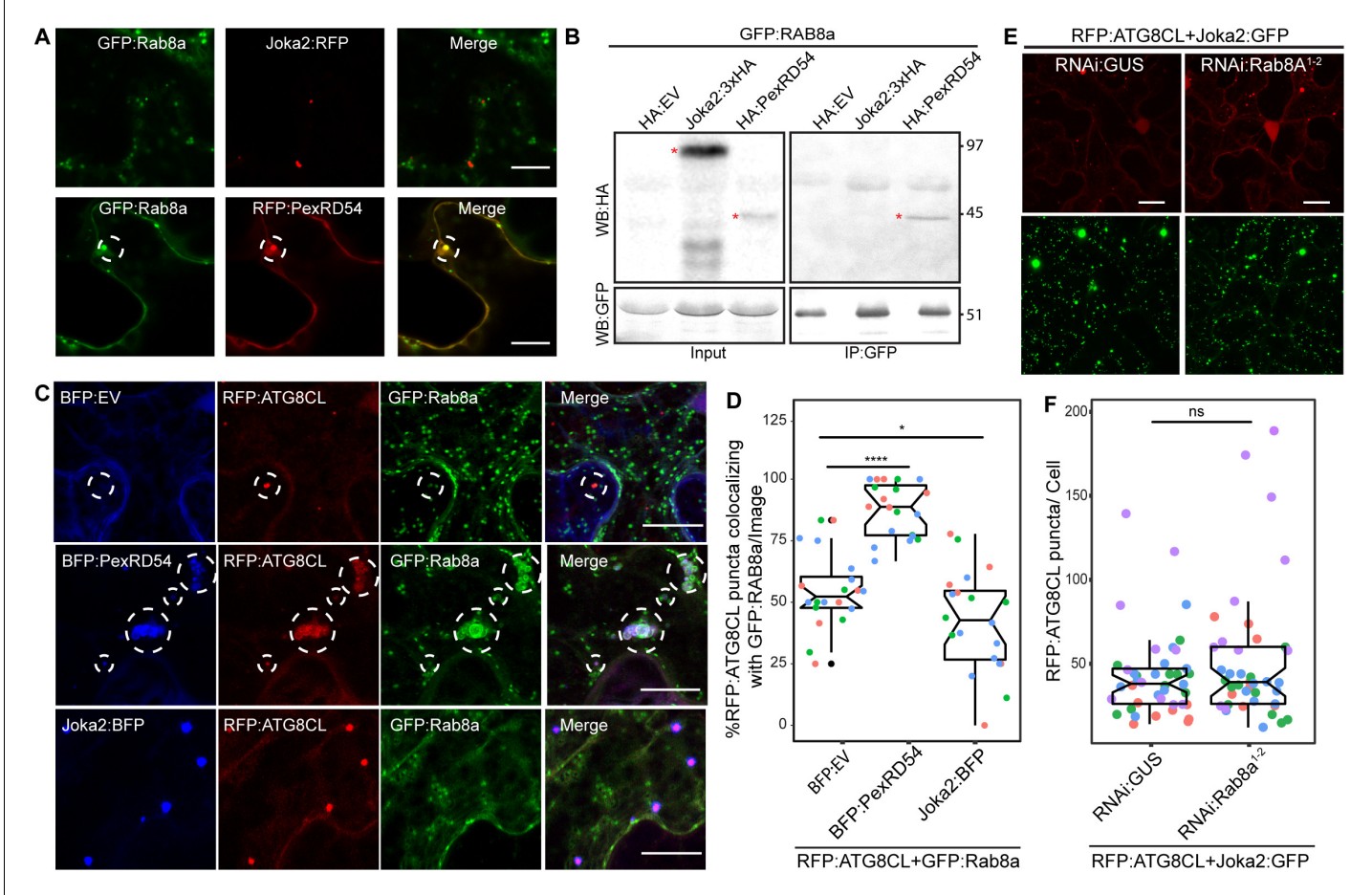

**Figure 6.** Rab8a is dispensable for Joka2-mediated autophagy. (**A**) Confocal micrographs of *Nicotiana benthamiana* leaf epidermal cells transiently expressing either Joka2:RFP (top) or RFP:PexRD54 (bottom), with GFP:Rab8a. Dashed circle highlights co-localized PexRD54 and Rab8a puncta. (**B**) In planta GFP pull-down assays of GFP:Rab8a and HA:EV, Joka2:HA or HA:PexRD54. Red asterisks indicate expected band sizes. (**C**) Maximum projection confocal micrographs of *N. benthamiana* leaf epidermal cells transiently expressing RFP:ATG8CL and Joka2:GFP, with either RNAi:GUS or RNAi: Rab8a[1-2]. Scale bars represent 10 µm. Images shown are maximal projections of 23 frames with 1 µm steps (**D**) Scatter-boxplot shows that silencing *NbRab8a1-2* (51, N = 41 images quantified) does not affect induction of ATG8CL autophagosome formation by Joka2 compared to a GUS silencing control (42, N = 42 images quantified). Statistical differences were analyzed by Students t-test in R. Measurements were significant when p<0.05 (*) and highly significant when p<0.0001(****). (**E**) Confocal micrographs of *N. benthamiana* leaf epidermal cells transiently expressing either Joka2:BFP (top), BFP:PexRD54 (middle) and BFP:EV (bottom), with RFP:ATG8CL and GFP:Rab8a. Scale bars represent 20 µm. Images shown are maximal projections of 16 frames with 1.2 µm steps (**F**) BFP:PexRD54 expression increases colocalization of RFP:ATG8CL and GFP:Rab8a puncta (88%, N = 20 images quantified) compared to Joka2:BFP (42%, N = 20 images quantified) and BFP:EV control (55% N = 20 images quantified), whereas Joka2:BFP slightly induces ATG8CL-Rab8a colocalization. Scattered points show individual data points, color indicates biological repeat. Statistical differences were analyzed by Welch Two Sample t-test in R. Measurements were significant when p<0.05 (*) and highly significant when p<0.0001(****).

The online version of this article includes the following source data and figure supplement(s) for figure 6:

**Source data 1.** Source data for western blots, autophagosome counts and colocalization values.

**Figure supplement 1.** Validation of *NbRab8a-1* silencing.

## PexRD54 triggers autophagy that is reminiscent of carbon-starvation-induced autophagy

Since we found that Joka2-mediated aggrephagy pathway does not necessarily rely on Rab8a (*Figure 6*), we decided to test whether other plant autophagy pathways employ Rab8a. Because autophagy can be induced through carbon starvation (*Huang et al., 2019*) and recent studies revealed a link between Rab8a, lipid droplets (LDs) and autophagy induced by carbon starvation in different systems (*Shpilka et al., 2015*; *Fan et al., 2019*; *Wu et al., 2014*), we tested whether Rab8a-ATG8CL association is altered during autophagy activation following light restriction. We detected a slight

yet significant increase in the number of RFP:ATG8CL puncta upon incubation of plants for 24 hr in the dark compared to normal light conditions (*Figure 7A–B*). Consistently, we detected slightly lower levels of endogenous ATG8s and transiently expressed RFP:ATG8CL following 24 hr dark treatment (*Figure 7—figure supplement 1*). However, when GFP:PexRD54 is present, we did not measure any further enhancement of RFP:ATG8CL puncta following 24 hr in the dark, suggesting

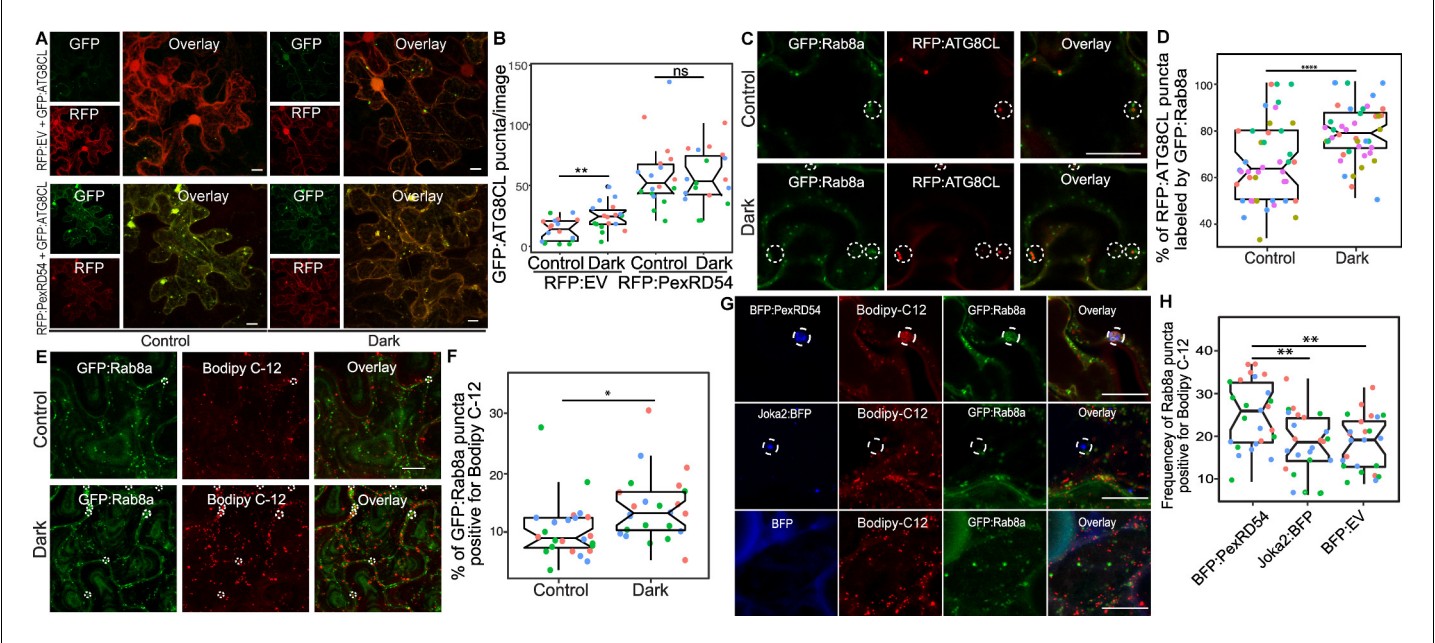

**Figure 7.** PexRD54 triggered autophagy is reminiscent of autophagy induced during carbon starvation. (**A**) Confocal micrographs of *N. benthamiana* leaf epidermal cells transiently expressing either RFP:EV or RFP:PexRD54, with GFPATG8CL under normal light or 24-hour-dark conditions. (**B**) Scatter-boxplot shows that dark treatment (24, *N* = 18 images quantified ) significantly increases RFP:ATG8CL-labeled puncta compared to control conditions (13, *N* = 18 images quantified ); however, when RFP:PexRD54 is present (59, *N* = 18 images quantified ), dark treatment does not further enhance puncta formation (57, *N* = 18 images quantified ). Images shown are maximal projections of 25 frames with 1 µm steps. (**C**) Colocalization of GFP:Rab8a and RFP:ATG8CL under normal light (top) or 24-hour-dark (bottom) conditions. Dashed circle shows co-localized ATG8CL and Rab8a puncta. (**D**) Dark treatment significantly increases percentage of RFP:ATG8CL puncta labeled by GFP:Rab8 (79%, *N* = 44 images quantified) compared to control conditions (66%, *N* = 44 images quantified). (**E**) Cells expressing GFP:Rab8a that are labeled by BODIPY-C$_{12}$ under normal light (top) and 24-hr-dark (bottom) conditions. Dashed circle shows co-localized Rab8a and BODIPY-C$_{12}$-positive puncta. (**F**) Twenty-four-hour-dark treatment increases the percentage of Rab8a puncta positive for BODIPY-C$_{12}$ (14%, *N* = 24 images quantified) compared to control conditions (10%, *N* = 24 images quantified). (**G**) Confocal micrographs of cells expressing either BFP:PexRD54 (top), Joka2:BFP (middle), or BFP:EV, with GFP:Rab8a and BODIPY C$_{12}$. Dashed circle in top panel shows BFP PexRD54 puncta is positively labeled by BODIPY-C$_{12}$ and GFP:Rab8a. Dashed circle in middle panel shows Joka2 puncta is negative for BODIPY-C$_{12}$ and GFP:Rab8a (**H**) Quantification of the puncta positive for both Rab8a and BODIPY C$_{12}$ shows enhanced frequency of colocalization by BFP:PexRD54 (25%, *N* = 25 images quantified) but not by Joka2:BFP (18%, *N* = 24 images quantified) or BFP:EV (18%, *N* = 24 images quantified). Scattered points show individual data points, color indicates biological repeat in panels B, D, F, and H. Statistical differences in panels B, D, and H were analyzed by Students t-test, and statistical differences in panel F were analyzed by Welch two sample t-test in R. Measurements were significant when p<0.05 (*) and highly significant when p<0.0001(****). Scale bars represent 10 µm.

The online version of this article includes the following source data and figure supplement(s) for figure 7:

**Source data 1.** Source data for western blots, autophagosome counts and colocalization values.

**Figure supplement 1.** Light restriction enhances depletion of both endogenous ATG8s and transiently expressed RFP:ATG8CL.

**Figure supplement 2.** Dark treatment increases ATG8CL/Rab8a colocalization.

**Figure supplement 3.** Rab8a puncta colocalize with Bodipy C12.

**Figure supplement 4.** Rab8a colocalizes in puncta and ring-shaped vesicle-like structures with ATG8CL and Bodipy C-12.

**Figure supplement 5.** PexRD54/Rab8a cluster localization with Bodipy C-12.

**Figure supplement 6.** PexRD54/Rab8a cluster localization with Bodipy C-12 and Oleosin.

**Figure supplement 7.** Stimulation of autophagy by PexRD54, but not Joka2, increases frequency of Rab8a and Bodipy C-12-positive puncta.

**Figure supplement 8.** Rab8a knockdown reduces the proportion of PexRD54 autophagosomes associated with Oleosin-labeled LDs.

**Figure supplement 9.** Validation of RNAi:Seipin silencing construct.

**Figure supplement 10.** Seipin knockdown reduces the number Oleosin-positive LDs.

**Figure supplement 11.** Seipin knockdown reduces the number of PexRD54 puncta.

that PexRD54-mediated autophagy can override or mask starvation-induced autophagy (**Figure 7A–B**). Furthermore, in plants stably expressing GFP:Rab8a that are light restricted, we detected an increased degree of colocalization between RFP:ATG8CL and GFP:Rab8a (**Figure 7C–D**, **Figure 7—figure supplement 2**), in a similar fashion to enhanced ATG8CL-Rab8a association mediated by PexRD54 (**Figure 4A–D**). Collectively, these data suggest that PexRD54 mimics carbon-starvation-induced autophagy.

Recent studies have revealed that lipid droplets (LDs) contribute to carbon-starvation-induced autophagy during light restriction (**Shpilka et al., 2015**; **Fan et al., 2019**). In addition, the GDP-bound mutant form of the mammalian Rab8a is enriched at LD contact sites to regulate their fusion (**Wu et al., 2014**). Therefore, we investigated Rab8a association with LDs under normal or starvation conditions. We first checked co-localization between GFP:Rab8a and LDs marked by the orange-red fluorescent fatty acid (FA), BODIPY 558/568 C12 (herein BODIPY-$C_{12}$). Confocal microscopy revealed that a small fraction of GFP:Rab8a puncta are labeled by the LD marker BODIPY-$C_{12}$ under normal light conditions, whereas the frequency of this colocalization increased by ~1.4-fold when plants are light restricted (**Figure 7E–F**, **Figure 7—figure supplement 3**). Furthermore, we observed that GFP:Rab8a puncta positive for BODIPY-$C_{12}$ are also labeled by BFP:ATG8CL (**Figure 7—figure supplement 4**). The stronger association of Rab8a and LDs upon light restriction, combined with the finding that LDs are recruited toward autophagosomes during carbon starvation (**Fan et al., 2019**), suggest that Rab8a-LD association could be a hallmark of starvation-induced autophagy.

Additionally, we observed that ring-shaped PexRD54 clusters labeled with Rab8a also tightly associate with BODIPY-$C_{12}$ labeled puncta (**Figure 7G**, **Figure 7—figure supplements 5–6**). However, we did not detect BODIPY-$C_{12}$ fluorescence signal filling the lumen of the PexRD54-labeled compartments (**Figure 7G**, **Figure 7—figure supplements 5–6**), indicating that fatty acids are likely not the autophagic cargoes of PexRD54. Rather, we detected a BODIPY-$C_{12}$ signal at the periphery of autophagosomes marked by PexRD54, which overlaps with GFP:Rab8a fluorescence signal (**Figure 7G**, **Figure 7—figure supplements 5–6**). Moreover, these PexRD54/Rab8a-clusters are also accompanied by LDs densely labeled with only BODIPY-$C_{12}$ as they navigate through the cytoplasm (**Video 3**). These findings suggest that fatty acids could be one of the potential membrane sources of the autophagosomes stimulated by PexRD54 as observed during carbon-starvation-induced autophagy in other systems (**Shpilka et al., 2015**). To gain further evidence for this, we investigated the colocalization of PexRD54 and Rab8a with the LD structural membrane protein Oleosin (**Siloto et al., 2006**; **Fan et al., 2019**; **Singh et al., 2009**). Similar to BODIPY-$C_{12}$-labeled puncta (**Figure 7—figure supplement 5**), Oleosin labeled LDs clustered around PexRD54/Rab8a-positive ring-like autophagosomes (**Figure 7—figure supplement 6**). Although Oleosin-positive LDs were adjacent to PexRD54 autophagosomes, in contrast to BODIPY-$C_{12}$, Oleosin-YFP did not produce fluorescent signal that overlaps with PexRD54/Rab8a ring-like autophagosomes (**Figure 7—figure supplement 6**), suggesting that FAs but not LD surface proteins are transferred to PexRD54-triggered autophagosomes. Strikingly, stimulation of autophagy by BFP:PexRD54, but not Joka2:BFP, led to enhanced association of BODIPY-$C_{12}$ and GFP:Rab8a-labeled puncta, supporting the hypothesis that PexRD54 mimics autophagy

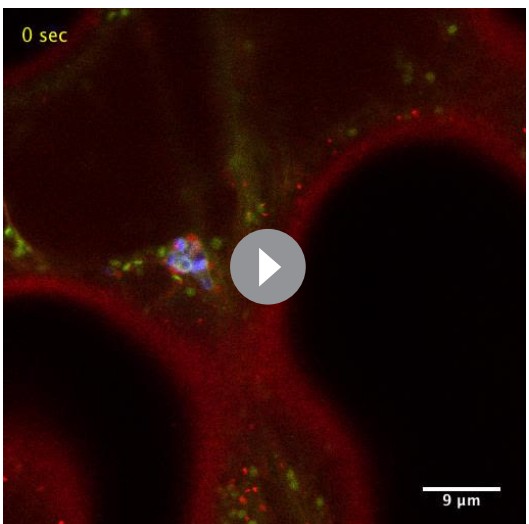

**Video 3.** Mobile PexRD54/Rab8a-positive puncta co-migrate with Bopidy-C12-labeled lipid droplets. GFP:Rab8a, BFP:PexRD54 are co-expressed three via agroinfiltration in *N. benthamiana* leaf epidermal cells stained with the lipid droplet dye Bodipy-C12. Confocal laser scanning microscopy was used to monitor PexRD54/Rab8a-labeled vesicles and lipid droplets 3 days post infiltration. The movie represents time-lapse of 56 frames acquired during 56 s (Frame interval: 1 s).
https://elifesciences.org/articles/65285#video3

induced during carbon starvation (*Figure 7H*, *Figure 7—figure supplement 7*). Together these results show that unlike the aggrephagy receptor Joka2, PexRD54 triggers responses similar to carbon-starvation-induced autophagy including induction of autophagosomes and enhanced association of ATG8CL-autophagosomes with Rab8a and LDs (*Figure 6* and *Figure 7*). However, it is possible that PexRD54 could activate other host autophagy pathways that favor pathogen virulence.

To investigate Rab8a's role in recruiting lipid droplets to PexRD54 puncta, we knocked down Rab8a with the RNAi:NbRab8a[1-4] construct and quantified the proportion of PexRD54 autophagosomes associated with Oleosin-labeled LDs. We observed that Rab8a knockdown not only reduces the total amount of PexRD54 autophagosomes but also the proportion of those associated with LDs compared to a silencing control (*Figure 7—figure supplement 8*). Finally to gain further evidence of the positive contribution of LDs to PexRD54-triggered autophagosomes, we designed a RNAi silencing construct to knockdown Seipin that has been shown to contribute to LD biogenesis in yeast, animals, and plants (*Cai et al., 2015*; *Yang et al., 2012*; *Taurino et al., 2018*; *Greer et al., 2020*). RNAi-mediated knockdown of the *NbSeipin-A* significantly reduced the amount of Oleosin-labeled LDs compared to a silencing control (*Figure 7—figure supplements 9–10*) consistent with the established role of Seipin in LD biogenesis. *NbSeipin-A* knockdown significantly reduced the amount of PexRD54 puncta in the cell compared to the control (*Figure 7—figure supplement 11*), supporting the notion that PexRD54 relies on host LD resources to stimulate autophagosome formation. Together, these data show that PexRD54 requires Rab8a to recruit lipid droplets and that LDs contribute to its induction of autophagosome formation. These findings are in agreement with the findings in yeast and mammalian cells that LDs supply membrane sources required for autophagosome formation during starvation-induced autophagy autophagosomes (*Shpilka et al., 2015*; *Dupont et al., 2014*).

## PexRD54 subverts Rab8a to autophagosomes at the pathogen interface

Our recent work revealed that the perihaustorial niche is a hot spot for the formation of ATG8CL autophagosomes stimulated by PexRD54 (*Dagdas et al., 2016*; *Dagdas et al., 2018*). Therefore, we next examined whether Rab8a-PexRD54 association occurs at perihaustorial ATG8CL-autophagosomes. We first checked GFP:Rab8a localization alone in the haustoriated cells. In infected leaf epidermal cells transiently or stably expressing GFP:Rab8a, we detected varying sizes of GFP:Rab8a puncta around the *P. infestans* haustoria (*Figure 8—figure supplement 1*). These structures included ring shaped compartments that are reminiscent of PexRD54-autophagosomes as well as smaller densely packed GFP-positive puncta and large vacuole like structures, indicating that Rab8a could regulate diverse trafficking pathways during infection (*Figure 8—figure supplement 1*, *Videos 4–5*). To verify that the perihaustorial Rab8a puncta represent the PexRD54-autophagosomes, we imaged infected plant cells which co-express GFP:Rab8a and the autophagosome marker protein RFP:ATG8CL in combination with BFP:PexRD54, BFP:AIMp, BFP, or Joka2:BFP. Confocal micrographs of haustoriated plant cells showed accumulation of RFP:ATG8CL-autophagosomes around the haustoria which are co-labeled with GFP:Rab8a, and are positive for BFP:PexRD54 but not BFP control (*Figure 8A–C*). However, formation of perihaustorial puncta

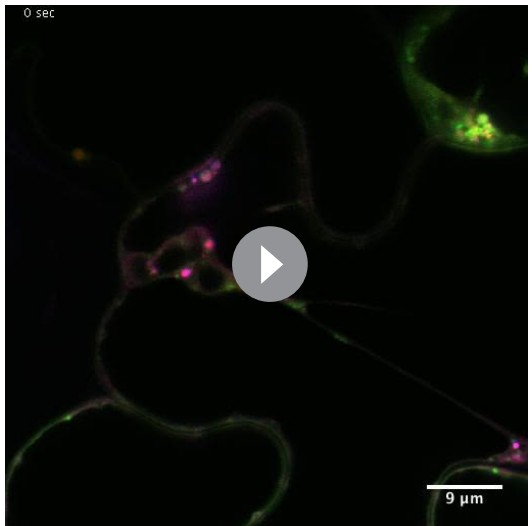

**Video 4.** Rab8a localizes to small mobile vesicles at haustoria during *P. infestans* infection. GFP:Rab8a, RFP:ATG8CL, and BFP:PexRD54 are co-expressed via agroinfiltration in *N. benthamiana* leaf epidermal cells infected with *P. infestans*. Confocal laser scanning microscopy was used to monitor Rab8a-labeled vesicles 3 days post infection. The movie represents time-lapse of 45 frames acquired during 3 min (Frame interval: 4 s).
https://elifesciences.org/articles/65285#video4

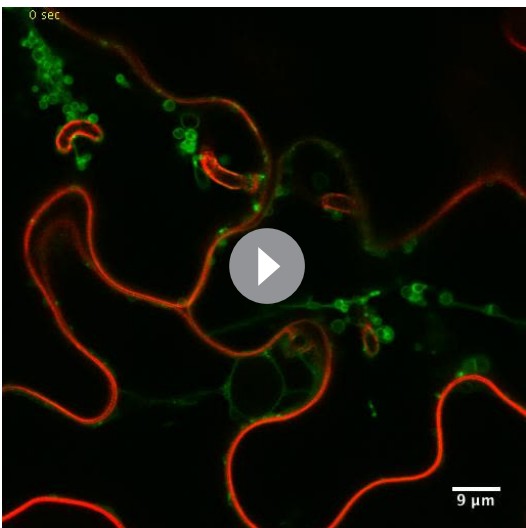

**Video 5.** Rab8a localizes to large vacuole-like structures at haustoria during *P. infestans* infection. GFP:Rab8a is co-expressed with the EHM marker RFP: REM1.3 via agroinfiltration in *N. benthamiana* leaf epidermal cells infected with *P. infestans*. Confocal laser scanning microscopy was used to monitor Rab8a-labelled vesicles 3 days post infection. The movie represents time-lapse of 26 frames acquired during 26 min (Frame interval: 30 s).

https://elifesciences.org/articles/65285#video5

co-labeled by RFP:ATG8CL and GFP:Rab8a was suppressed by arresting autophagosome formation through expression of BFP:AIMp (*Figure 8B*). Notably, we detected a subset of perihaustorial GFP:Rab8a puncta that are not labeled by RFP:ATG8CL, further supporting an ATG8CL-independent haustorial trafficking role for Rab8a (*Figure 8A–D*, *Figure 8—figure supplement 2*). In agreement with these findings, other Rab8 members including RabE1 family have been reported to localize to distinct subcellular compartments including Golgi and Peroxisomes in *Arabidopsis* (*Zheng et al., 2005*; *Cui et al., 2013*). Therefore, we next investigated the potential colocalization of Rab8a and PexRD54 with markers that label organelles such as Golgi, peroxisomes and mitochondria. Co-expression of Rab8a and PexRD54 with GmMan1$_{1-49}$-mCherry (Golgi marker), ScCOX4$_{1-29}$-mCherry (mitochondria marker), or CFP-GLOX (peroxisome marker) revealed extensive co-localization of Rab8a with the Golgi marker but not with the markers that label mitochondria or peroxisomes (*Figure 8—figure supplement 3A*; *Nelson et al., 2007*). However, PexRD54-Rab8a puncta did not show any labeling by any of the organelle markers tested (*Figure 8—figure supplement 3A*). Consistent with these findings, Rab8a distinctly localized at either the Golgi or PexRD54 puncta, but not at peroxisomes or mitochondria, at the haustorium interface (*Figure 8—figure supplement 3B*). Therefore, we conclude that Rab8a localization to PexRD54 autophagosomes cannot be explained by Rab8a's association with these organelles. Intriguingly, we noticed that peroxisomes frequently appear around the PexRD54-Rab8a puncta, suggesting a potential involvement of this organelle in PexRD54-mediated autophagy (*Figure 8—figure supplement 3*).

We also observed that Rab8a and PexRD54 co-localize with both Bodipy C-12 and oleosin in clusters of vesicles around haustoria, linking Rab8a's emerging role in lipid trafficking with the pathogen effector and haustorial interface (*Figure 8—figure supplement 4*). On the other hand, in line with our findings that the Joka2 pathway does not employ Rab8a (*Figure 6*), perihaustorial Joka2:BFP/RFP:ATG8CL puncta and GFP:Rab8a puncta were exclusive to each other (*Figure 8D*). Consistent with our pull down assays (*Figure 3D*), we did not detect any sharp GFP:Rab8a$^{Q74L}$ signal at the perihaustorial BFP:PexRD54 puncta (*Figure 8—figure supplement 5*). In contrast, GFP:Rab8a, and particularly GFP:Rab8a$^{S29N}$, produced strong fluorescence signals peaking at perihaustorial BFP: PexRD54 puncta (*Figure 8—figure supplement 5*), indicating that both wild-type Rab8a and Rab8a$^{S29N}$ are enriched at the perihaustorial PexRD54 autophagosomes. Taken together, these results demonstrate that Rab8a localizes to distinct compartments that accumulate around the haustorium and PexRD54 stimulates diversion of Rab8a positive LDs to perihaustorial autophagosomes.

Localization of Rab8a to Golgi around the haustorium is in agreement with the conserved role of Rab8 in yeast and metazoans to mediate polarized secretion of proteins (*Nielsen et al., 2008*). These findings, combined with our data that PexRD54-mediated subversion of Rab8a to autophagosomes (*Figure 8*) around the haustoria, imply a potential function of Rab8a in polarized defense-responses. Therefore, we next investigated the possible role of Rab8a in immunity against *P. infestans*. Simultaneous RNAi-mediated knockdown of all four *NbRab8a* members led to a consistent increase in disease symptoms and *P. infestans* hyphal growth (*Figure 8—figure supplement 6*). To further determine the role of the Rab8a family in immunity, we conducted a silencing complementation assays using a codon shuffled *NbRab8a-1* construct fused to GFP

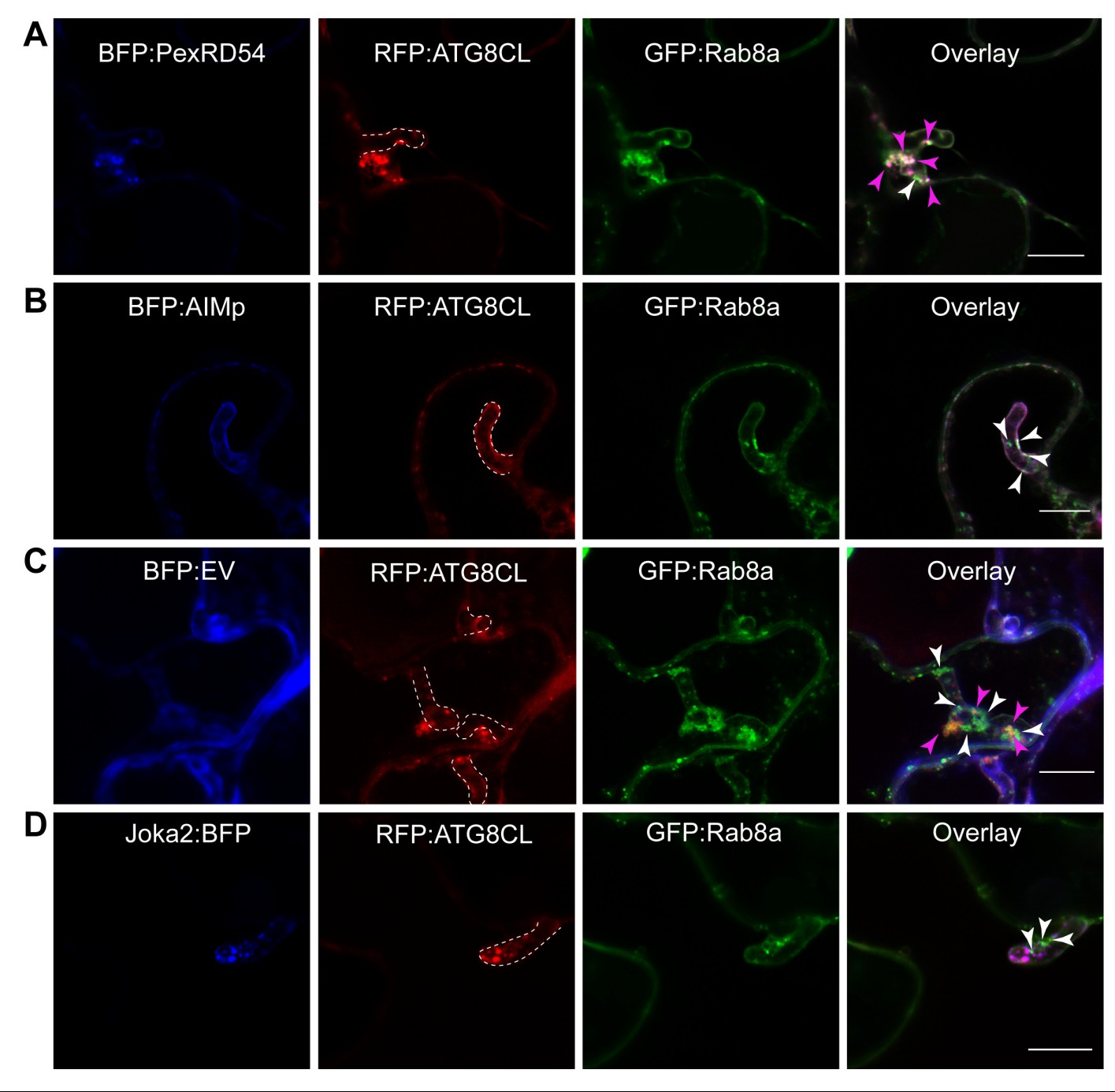

**Figure 8.** Rab8a is recruited to perihaustorial PexRD54-ATG8CL autophagosomes. (**A-D**) Confocal micrographs of *P. infestans*-infected *N. benthamiana* leaf epidermal cells transiently expressing either BFP:PexRD54 (**A**), BFP:EV (**B**), BFP:AIMp (**C**) or Joka2:BFP (**D**), with both RFP:ATG8CL and GFP:Rab8a. BFP:PexRD54 co-localizes with RFP:ATG8CL and GFP:Rab8a at perihaustorial region, whereas Joka2:BFP-labeled ATG8CL puncta are exclusive to GFP:Rab8a puncta. Haustoria are labeled with white dashed lines. Pink arrowheads highlight vesicles labeled by ATG8CL and Rab8a, whereas white arrowheads highlight vesicles labeled by Rab8a only.

The online version of this article includes the following source data and figure supplement(s) for figure 8:

**Source data 1.** Source data for infection assays.

**Figure supplement 1.** Rab8a localizes to various vesicle-like structures around haustoria during infection with *P. infestans*.

**Figure supplement 2.** Two distinct populations of Rab8a vesicles localize to *P. infestans* haustoria.

**Figure supplement 3.** Rab8a localizes to PexRD54-autophagosomes or Golgi, but not to mitochondria or peroxisomes.

*Figure 8 continued on next page*

*Figure 8 continued*

**Figure supplement 4.** Rab8a colocalizes with PexRD54, Oleosin and Bodipy C-12, but not with the mitochondria marker, in puncta around *P. infestans* haustoria.

**Figure supplement 5.** Localisation of Rab8a and its mutants in perihaustorial autophagosomes during infection by *P. infestans*.

**Figure supplement 6.** Rab8a positively contributes to immunity to *P. infestans*.

**Figure supplement 7.** Silencing resilient synthetic Rab8a (Rab8a$^{syn}$) rescues enhanced pathogen susceptibility phenotype caused by NB:*Rab8a1-4* silencing.

**Figure supplement 8.** Overexpression of Rab8a$^{N128I}$ significantly increases *P*.

(GFP:Rab8a$^{syn}$) that can evade RNAi (*Figure 8—figure supplement 7*). The enhanced susceptibility phenotype caused by RNAi:NbRab8a$^{1-4}$ silencing was rescued by simultaneous overexpression of GFP:Rab8a$^{syn}$ but not the GFP control, providing further evidence that Rab8a is required for basal resistance against *P. infestans* (*Figure 8—figure supplement 7*). Consistent with knockdown assays, overexpression of dominant negative GFP:Rab8a$^{N128I}$ enhanced plant susceptibility to *P. infestans* compared to a GFP control (*Figure 8—figure supplement 8*). Collectively, these results indicate that Rab8a contributes to plant immunity, whereas PexRD54 could possibly interfere with the defense-related role of Rab8a by subverting a subpopulation of it to autophagic compartments.

## AIM peptide-mediated autophagy arrest leads to reduced pathogen growth

The observed induction of haustoria targeted autophagosomes prompted the hypothesis that *P. infestans* could benefit by co-opting host autophagy to support its own growth. Therefore, we next explored how activation of autophagy by PexRD54 affects *P. infestans* host colonization. Our discovery of the AIMp as an ATG8-specific autophagy inhibitor that can be used to spatiotemporally arrest plant autophagy allowed us to test the impact of autophagy on *P. infestans* virulence. To this end, we decided to transiently interfere with pathogen-induced autophagy by expressing the AIMp. Our quantitative image analysis revealed that compared to RFP:PexRD54 expression, transient expression of RFP:AIMp led to ~threefold decrease in the number of haustoria that are associated with autophagosomes marked by GFP:ATG8CL (*Figure 9A–B*). We then measured how autophagy suppression by the AIMp affects *P. infestans* infection. In multiple independent experiments (six biological replicates), *N. benthamiana* leaf patches expressing RFP: AIMp showed a consistent reduction of quantitative disease symptoms compared to an RFP vector control (*Figure 9C–D*). This indicated that AIMp-mediated arrest of host autophagy negatively impacts *P. infestans* virulence, supporting the hypothesis that PexRD54-triggered autophagy is beneficial to the pathogen. Collectively, these results suggest that *P. infestans* relies on host autophagy function to support its virulence. This could explain why the pathogen deploys full-length PexRD54 that can activate specific host autophagy pathways while subverting defense-related autophagy, instead of just the AIM peptide.

## Discussion

Dissecting the specialized functions and mechanisms of autophagy in host-microbe interactions has been challenging. This is mainly due to prolonged stress accumulation in autophagy mutants and non-autophagy related roles of the targeted genes (*Munch et al., 2014*). Pathogen effectors that target specific components of the host autophagy machinery have emerged as alternative tools to unravel the underlying mechanisms of defense-related autophagy (*Dagdas et al., 2016*; *Hafrén et al., 2018*; *Üstün et al., 2018*). Here, by studying the *P. infestans* effector protein PexRD54, we shed light on the poorly understood mechanism of pathogen-induced autophagy in plants. We unveil a distinct pathogen virulence mechanism in which the effector protein couples host vesicle transport machinery to autophagosome biogenesis. Through its modular domain architecture, PexRD54 employs a diverse set of host proteins such as Rab8a and ATG8CL to stimulate autophagy that is reminiscent of starvation-induced autophagy. We propose a model in which PexRD54 recruits Rab8 and lipid droplets to the pathogen feeding site to stimulate autophagosome biogenesis (*Figure 10*). Both PexRD54 and carbon starvation drive association of

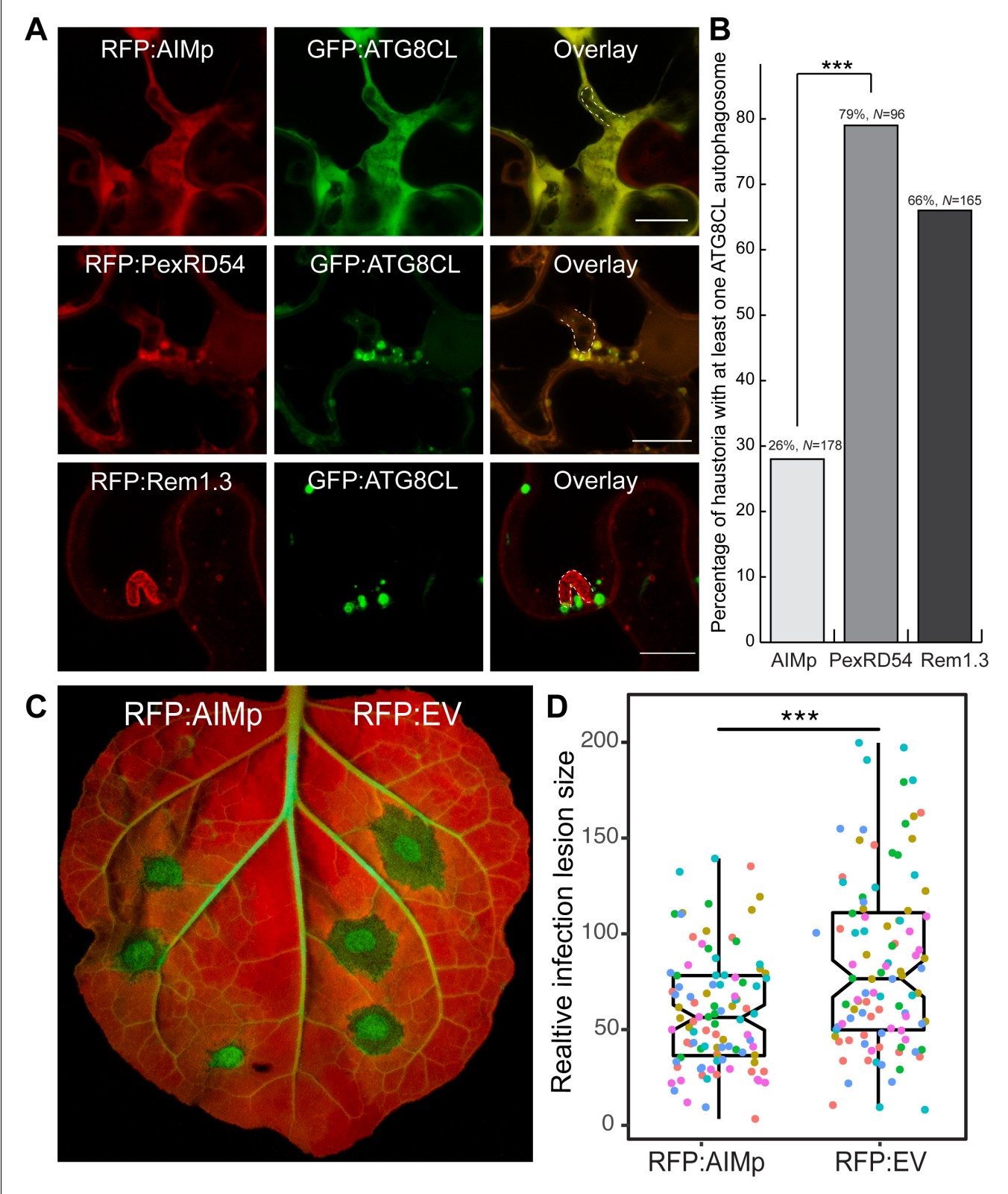

**Figure 9.** AIM peptide mediated arrest of ATG8 autophagy negatively affects *P. infestans* infection. (**A**) Confocal micrographs of *P. infestans*-infected *N. benthamiana* leaf epidermal cells transiently expressing either RFP:AIMp (top), RFP:PexRD54 (middle), or PM-haustorial marker RFP:Rem1.3 (bottom), with GFP:ATG8CL. Haustoria are labeled with white dashed lines. (**B**) Co-expressing RFP:AIMp with GFP:ATG8CL substantially decreases the percentage of haustoria associated with ATG8CL-labeled puncta (26%, *N* = 178 haustoria), compared to PexRD54 (79%, *N* = 96 haustoria) and

*Figure 9 continued on next page*

*Figure 9 continued*

haustorial marker control RFP:Rem1.3 (66%, *N* = 165 haustoria). Statistical differences were analyzed by Pearson's chi-squared test in R. Measurements were significant when p<0.05 (*) and highly significant when p<0.001(***). (C) AIMp reduces disease symptoms of *P. infestans* (65, *N* = 84 infected leaves) compared to empty vector control (87, *N* = 85 infected leaves). *N. benthamiana* leaves expressing RFP:AIMp and RFP:EV were infected with *P. infestans* and pathogen growth was determined by measuring infection lesion size 7 days post-inoculation. (D) Box plot shows relative infection lesion size of 84 and 85 infection sites from five biological replicates, respectively. Scattered points indicate individual data points and different colors represent various biological repeats. Statistical differences were analyzed by Welch Two Sample t-test in R. Measurements were significant when p<0.05 (*) and highly significant when p<0.001(***).

The online version of this article includes the following source data for figure 9:

**Source data 1.** Source data for infection assays.

Rab8a and LDs which are positioned at autophagy compartments marked by ATG8CL. We hypothesize that the pathogen could benefit from this process potentially through subverting the immune function of Rab8a and gaining access to plant resources carried in autophagosomes diverted to haustoria.

## Autophagy suppression by the AIM peptide

Effectors are excellent tools to dissect complex biological processes such as autophagy as they often display high target specificity (*Schardon et al., 2016*). In this study, we not only expanded our knowledge of pathogen-triggered autophagy but also discovered an effector derived peptide (AIMp) that can specifically block autophagy. Here, the AIMp served as an excellent negative control to understand PexRd54 activities and to interpret the impact of autophagy manipulation by the pathogen. We used it as a tool to perturb PexRD54 activities without directly interfering with Rab8a functions. Most *ATG* mutants show pleiotropic affects and have non-autophagy-related roles, making it difficult to interpret the outcomes of autophagy inhibition. However, the AIMp blocks autophagy specifically, as it directly acts on ATG8. Chemical inhibitors are also often used to measure autophagy flux. However, these inhibitors are mostly inefficient and lack the required specificity. We discovered that PexRD54's AIM peptide is a strong autophagy suppressor effective against all potato ATG8 isoforms (*Figure 2*). Our data suggest that AIMp blocks the autophagosome biogenesis step, as both autophagosome formation and autophagic flux are suppressed by the AIMp (*Figures 1–2*). Conceivably, the AIM peptide competitively inhibits autophagosome biogenesis by occupying the AIM docking site of ATG8 that accommodates host autophagy regulators (*Noda et al., 2008*). In agreement with our finding, autophagy cargo receptors that bind ATG8 via AIMs were recently shown to stimulate ATG8 (*Chang et al., 2021*). The AIM peptide is a genetically encodable tool, which can enable spatio-temporal arrest of autophagy when expressed under inducible or tissue specific promoters. Thus, this should be of great interest for autophagy studies in plants and other systems, which can overcome the limitations of chemical autophagy inhibitors and autophagy mutants. We also developed AIM peptide derivatives with cell penetrating features, which allow studying the tissue-specific functions of autophagy. The cell penetrating AIM peptide can be used to study autophagy in plants and other eukaryotic systems that are not amenable to genetic manipulation.

## How does PexRD54 activate autophagy?

Autophagosome biogenesis is a complex multi-step process. But how could an effector activate such an intricate process? We uncovered that PexRD54 either directly or indirectly recruits Rab8a to autophagosome biogenesis sites (*Figures 3–4*). Similar channeling of Rab8a to ATG8CL-autophagosomes occurs during carbon starvation (*Figure 7*), suggesting that PexRD54 mimics autophagy induction via nutrient deprivation. Interestingly, the mammalian autophagy cargo receptor Optineurin also interacts with both LC3 (mammalian ATG8 isoform) and Rab8a in mice (*Bansal et al., 2018*; *Vaibhava et al., 2012*). Although the functional implications of these interactions are unclear, a proposed model suggests that Optineurin mediates pre-autophagosomal membrane elongation through anchoring Rab8a to autophagosome assembly sites. Our data is consistent with this model and suggests that PexRD54 recruits Rab8a to facilitate autophagosome formation.

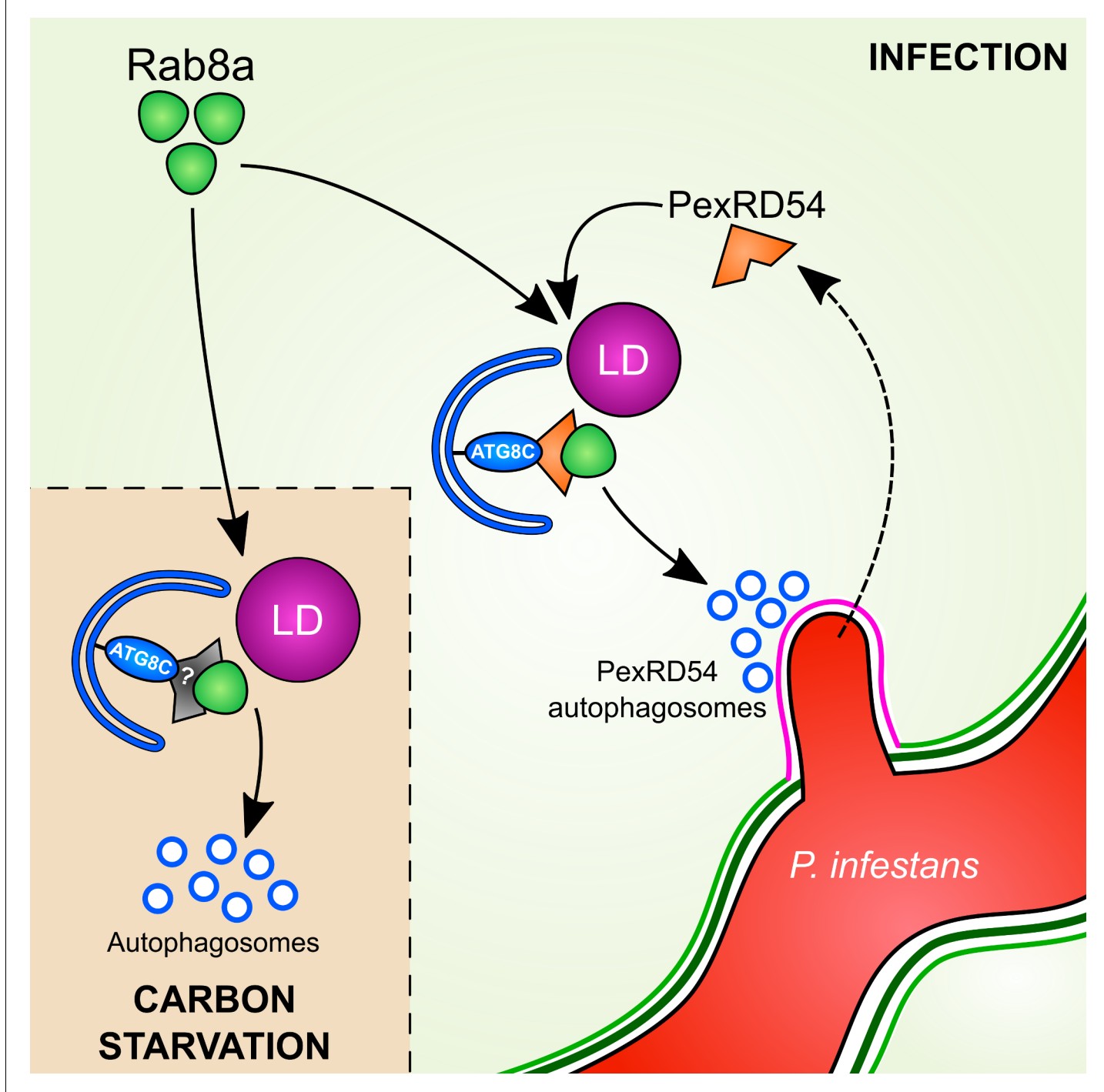

**Figure 10.** Model for PexRD54 subversion of host vesicle trafficking to stimulate autophagosome biogenesis at the haustorial interface. Under carbon starvation conditions, Rab8a recruits lipid droplets (LDs) to the phagophore assembly site that contains ATG8CL to drive starvation-induced autophagy, possibly via an unknown cargo receptor or adaptor protein. During *P. infestans* infection, the RXLR effector PexRD54 is translocated inside the host cells across the extra-haustorial membrane (magenta). Inside the host cells PexRD54 interacts with the host autophagy protein ATG8CL and co-opts Rab8a to recruit LDs and induce the formation of autophagosomes which are directed towards the haustoria.

## The role of Rab8a in autophagy

The membrane elongation step of autophagosome formation relies on direct transport of lipids from various donor compartments. LDs provide a membrane source for autophagosome biogenesis specifically during starvation-induced autophagy (*Shpilka et al., 2015*). More recently, a conserved acyl-CoA synthetase (ACS) from yeast was shown to be mobilized on nucleated phagophores where it locally mediates transport of fatty acids required for phagophore elongation (*Schütter et al., 2020*). But how are these lipid sources mobilized to autophagosome biogenesis sites at first? Intriguingly, the GDP bound form of the mammalian Rab8a mediates LD fusion events (*Wu et al., 2014*). In line with these reports, we found that PexRD54 associated more strongly with the Rab8a$^{S29N}$ mutant (presumably the GDP bound form) and recruits Rab8a to autophagosome biogenesis sites to enhance LD localization around the ATG8CL-foci and stimulate autophagosome formation (*Figures 4* and *7*, *Figure 4—figure supplement 3*, *Figure 7—figure supplements 4–10*). However, we did not find any luminal signal with the FA marker, indicating that FAs are not the likely the cargoes of PexRD54. Furthermore, we detected FAs labeled with Bodipy but not the LD membrane marker protein Oleosin localized to periphery of the PexRD54 compartments (*Figure 7G*, *Figure 7—figure supplements 5–6*). This suggests that PexRD54 could employ Rab8a to position LDs at ATG8CL nucleation sites to facilitate lipid transfer for autophagosome biogenesis. Interestingly, PexRD54-triggered association of Rab8a and LDs is also enhanced by carbon starvation but not by Joka2 (*Figure 7G–H*, *Figure 7—figure supplement 6*). We propose that upon carbon starvation, the plant uses LDs as an additional membrane source to accommodate for an increase in autophagosome biogenesis, and PexRD54 exploits this process to stimulate autophagy. Nevertheless, Rab8a may not be engaged in all autophagy routes, as it is dispensable for Joka2-mediated autophagy (*Figure 6*). This is consistent with the finding that Arabidopsis NBR1 (Joka2) mutants are sensitive to a variety of abiotic stress conditions but not to carbon starvation (*Zhou et al., 2013*). Our results are consistent with the recent findings in yeast, where LDs specifically contribute to starvation-induced autophagy (*Shpilka et al., 2015*). Nevertheless, further assays are needed to determine whether Rab8a is implicated in other autophagy pathways. Altogether, combined with previous findings, we conclude that distinct cellular transport pathways feed autophagosome formation during diverse selective autophagy pathways in plants.

## The role of Rab8a in immunity

Our data revealed that the Rab8a family contributes to basal resistance against *P. infestans* (*Figure 8—figure supplements 6–8*). Rab8a belongs to the Rab8 family of small GTPases that are implicated in polarized secretion events in eukaryotes (*Pfeffer, 2017*). Another Rab8 member known as RabE1d was found to contribute to bacterial resistance and regulate membrane trafficking in the model plant Arabidopsis (*Speth et al., 2009*; *Zheng et al., 2005*). However, the extent to which Rab8 family members function in immunity remains unknown. Our findings revealed that Rab8a is most likely involved in a diverse range of cellular transport pathways including autophagy. Consistent with the evolutionarily conserved role of Rab8 family members in polarized secretion, we detected Rab8a localization to Golgi around the haustorium. These findings suggest that Rab8a has a function in defense-related secretion that possibly contributes to plant focal immune responses. Further research is required to determine the cargoes and the potential defense-related functions of Rab8a during infection.

## Why does *P. infestans* activate autophagy?

Autophagy suppression by the AIM peptide leads to reduced pathogen virulence (*Figure 9*), indicating that host autophagy inhibition is not favorable to *P. infestans*. This could explain why *P. infestans* deploys PexRD54, which can neutralize defense-related autophagy mediated by Joka2, while enabling other autophagy pathways that are somewhat beneficial. Intriguingly, the autophagy pathway primed by PexRD54 resembles autophagy induced by carbon starvation, but not to autophagy induced by the aggrephagy cargo receptor Joka2. In contrast to Joka2, both PexRD54 and light restriction stimulated enhanced Rab8a-ATG8CL and Rab8a-LD associations (*Figure 6* and *Figure 7*). Furthermore, their effect on autophagosome formation were not additive (*Figure 7A–B*). This hints at the possibility that PexRD54 could facilitate nutrient uptake from the host cells by mimicking starvation conditions to stimulate autophagy. This view is further supported by our earlier finding that

PexRD54-autophagosomes are diverted to the haustorial interface (*Dagdas et al., 2016*). However, our data suggests that the fate of PexRD54 autophagosomes in infected and uninfected cells could be different, since in uninfected cells PexRD54 is degraded inside the plant vacuole following 4–6 days of ectopic expression (*Figure 2—figure supplements 1–3*). Therefore, PexRD54 may or may not be degraded in infected plant cells. PexRD54 is a member of RXLR effectors which typically show high level of expression during 2–5 days of infection. Consequently, even if PexRD54 is depleted via host autophagy machinery during infection, there would be more of it secreted via the haustorium to maintain PexRD54 virulence functions as long as the haustorium is accommodated in the host cells. PexRD54 likely remodels the cargoes engulfed in ATG8CL-autophagosomes targeted to pathogen interface, which are presumably assimilated by the parasite. Whether these cargoes are hydrolyzed in infected plant cells or directly absorbed by the pathogen remains to be determined. Alternatively, but not mutually exclusively, PexRD54 could also help neutralize defense-related host components by engulfing them in secure membrane-bound autophagy compartments. For instance, PexRD54 could promote diversion of Rab8a to autophagy to intervene with non-autophagy related immune functions of Rab8a (*Figure 8*). Therefore, we cannot exclude the possibility that PexRD54 could activate other host autophagy pathways that are beneficial to the pathogen, and further research is needed to determine whether PexRD54 could mimic other autophagy inducing conditions.

In summary, our findings demonstrate that to support its virulence, *P. infestans* manipulates plant cellular degradative and transport systems by deploying an effector protein that imitates carbon starvation conditions. It also demonstrates effectors can act as adaptors to bridge multiple host components to modulate complex cellular processes for the benefit the pathogen. Further research is needed (i) to determine the cargo of these autophagosomes, (ii) whether they are secreted to the pathogen interface and subsequently absorbed by the pathogen, and (iii) the molecular players involved in pathogen subverted autophagy.

# Materials and methods

## Key resources table

| Reagent type (species) or resource | Designation | Source or reference | Identifiers | Additional information |
|---|---|---|---|---|
| Gene (*Solanum Tuberosum*) | StRab8a (Rab8a) | Sol Genomics Network | Sotub04g010260.1 | |
| Gene (*Nicotiana benthamiana*) | NbRab8a-1 | Sol Genomics Network | Niben101Scf 07650g01021 | |
| Gene (*Nicotiana benthamiana*) | NbRab8a-2 | Sol Genomics Network | Niben101Scf 09596g00001 | |
| Gene (*Nicotiana benthamiana*) | NbRab8a-3 | Sol Genomics Network | Niben101Scf00684g00002 | |
| Gene (*Nicotiana benthamiana*) | NbRab8a-4 | Sol Genomics Network | Niben101Scf03277g02014 | |
| Gene (*Nicotiana benthamiana*) | NbSeipin-A | Sol Genomics Network | Niben101Scf01983g13002 | |
| Gene (*Nicotiana benthamiana*) | NbSeipin-B | Sol Genomics Network | Niben101Scf03695g01004 | |
| Biological sample (*Nicotiana benthamiana*) | Nb-GFP-StRab8a seeds | This paper | | Seeds are maintained at Bozkurt lab (ICL) |

*Continued on next page*

*Continued*

| Reagent type (species) or resource | Designation | Source or reference | Identifiers | Additional information |
|---|---|---|---|---|
| Antibody | Anti-GFP (Rabbit polyclonal) | Chromotek | Cat# PABG1-100 RRID:AB_2749857 | WB: 1:1000 |
| Antibody | Anti-HA (Rat monoclonal) | Chromotek | Cat# 7c9-100 RRID:AB_2827568 | WB: 1:1000 |
| Antibody | Anti-RFP (Rat monoclonal) | Chromotek | Cat# 6g6-100 RRID:AB_2631395 | WB: 1:1000 |
| Antibody | Anti-FLAG (Mouse monoclonal) | Sigma-Aldrich | Cat# F3165 RRID:AB_259529 | WB: 1 μg/mL |
| Antibody | Anti-tRFP (Mouse monoclonal) | Evrogen | Cat# AB233, RRID:AB_2571743 | WB: 1:1000 |
| Peptide, recombinant protein | AIMp$^{Syn}$ | This paper | Peptide Sequence | RKKRRRESRKKRR RESKPLDFDWEIV |
| Peptide, recombinant protein | mAIMp$^{SynAA}$ | This paper | Peptide Sequence | RKKRRRESRKKRRR ESKPLDFDAEIA |
| Sequence-based reagent | GA_RD54_F | This paper | PCR Primers | CTGGATCTGGAGAATTT GATGTTGGTCCCTCTTGGCT |
| Sequence-based reagent | GA_RD54_R | This paper | PCR Primers | TAGCATGGCCGCGGGATTT ACACAATTTCCCAGTCG |
| Sequence-based reagent | GA_LIR2_R | This paper | PCR Primers | TAGCATGGCCGCGGGATTT AAGCAATTTCCGCGTCG |
| Sequence-based reagent | GA_AIMp_F | This paper | PCR Primers | CTGGATCTGGA GAATTTGATCG GGACAAAATTGACAAGA |
| Sequence-based reagent | GA_ATG8C_F | This paper | PCR Primers | CTGGATCTGGA GAATTTGATG CCAAAAGCTCCTTCAAA |
| Sequence-based reagent | GA_ATG8C_R | This paper | PCR Primers | TAGCATGGCCGCGGGATT CAAAAGGATCCGAAGGTAT |
| Sequence-based reagent | GAJoka2BFP_F | This paper | PCR Primers | CAGGCGGCCGCACTAGTGAT ATGGCTATGGAGTCAT CTATTGTGATCAAGG |
| Sequence-based reagent | GAJoka2BFP_R | This paper | PCR Primers | GCAGATCCAGCAGA TCCGATCTGCT CTCCAGCAATAAGA TCCATCACAAC |
| Sequence-based reagent | NbRab8A_silF1 | This paper | PCR Primers | ACCAGGTCTCAGGAGGGC TTATATAAATGAAGCGAC |
| Sequence-based reagent | NbRab8A_silR1 | This paper | PCR Primers | ACCAGGTCTCAT CGTACTTCTG CAATCGCGTGCGT CCGAAGGTAT |
| Sequence-based reagent | GW_StRab8a-1_F | This paper | PCR Primers | CACCATGGCCGCTCCA CCCGCTAGAGCTCGAGCT |
| Sequence-based reagent | GW_StRab8a-1_R | This paper | PCR Primers | TTAAGAACCACAGCAAG CTGATTTTTGGGCG |
| Sequence-based reagent | Rab8aS29N_F | This paper | PCR Primers | GTTCTTACCCACAC CGCTGTCGCCG |
| Sequence-based reagent | Rab8aS29N R | This paper | PCR Primers | TGCCTTCTTTTACGTT TCTCAGATG |
| Sequence-based reagent | Rab8aQ74L_F | This paper | PCR Primers | AAACCGGCGGTATCC CAGATTTGCAG |

*Continued on next page*

*Continued*

| Reagent type (species) or resource | Designation | Source or reference | Identifiers | Additional information |
|---|---|---|---|---|
| Sequence-based reagent | Rab8aQ74L_R | This paper | PCR Primers | GGAGCGGTTCCGA ACAATTACAACT |
| Sequence-based reagent | Rab8aN128I_F | This paper | PCR Primers | ATGCCGACCAGAATT TTGTTGACATTG |
| Sequence-based reagent | Rab8aN128I_R | This paper | PCR Primers | CAAGGCTGACATGG ATGAAAGCAAAAGG |
| Sequence-based reagent | Rab8a1-4-RNAi_F1 | This paper | PCR Primers | ACCAGGTCTCAGGAGGC AGCTCCACCAGCTAGG |
| Sequence-based reagent | Rab8a1-4-RNAi_R1 | This paper | PCR Primers | ACCAGGTCTCAGTGAAGG AACCATCTGAGAAC |
| Sequence-based reagent | Rab8a1-4-RNAi_F2 | This paper | PCR Primers | ACCAGGTCTCATCACACC ACTATTGGTATTGAT |
| Sequence-based reagent | Rab8a1-4-RNAi_R2 | This paper | PCR Primers | ACCAGGTCTCAATTGTT CGGAAACGCTCCTGG |
| Sequence-based reagent | Rab8a1-4-RNAi_F3 | This paper | PCR Primers | ACCAGGTCTCACAATCAA GATAAGGACCATTGAGT |
| Sequence-based reagent | Rab8a1-4-RNAi_R3 | This paper | PCR Primers | ACCAGGTCTCATCGTCAT GTCAGCCTTGTTGCCG |
| Sequence-based reagent | NYFP-Oleosin F | This paper | PCR Primers | GCAGAAGGTTATGA ACCACGATGCA GATTACTATGGGC AGCAACATAC |
| Sequence-based reagent | Oleosin-CYFP R | This paper | PCR Primers | TCTGCTTAACCATG TTGTGGATAC TCTGCTGGGTTC CAGTGACATG |
| Sequence-based reagent | GA_35S_HA_F | This paper | PCR Primers | GCCGCACTAGTG ATATGTACCCA |
| Sequence-based reagent | GA_Term_R | This paper | PCR Primers | ATTTTTGCGGACT CTAGCATGG |
| Sequence-based reagent | GAPDH_F | This paper | PCR Primers | ATGGCTTCTCAT GCAGCTTT |
| Sequence-based reagent | GAPDH_R | This paper | PCR Primers | ATCCTGTGGTCT TGGGAGTG |
| Sequence-based reagent | RFP F: | This paper | PCR Primers | CTGGACATCACCT CCCACAACGAGG |
| Sequence-based reagent | Term R: | This paper | PCR Primers | CACATGAGCGAAACC CTATAAGAACCCTA |
| Sequence-based reagent | GLOX- F | This paper | PCR Primers | CACCATGGCGGAGAC GGTCACCAATGTAT |
| Sequence-based reagent | GLOX- R | This paper | PCR Primers | TTACATCCTCGGCAGGGG |

## Plant material and growth conditions

*N. benthamiana* WT and transgenic plants (35S::GFP:Rab8a and 35S::GFP:ATG8CL) were grown and maintained in a greenhouse with high light intensity (16 hr light/8 hr dark photoperiod) at 22–24°C. To apply carbon stress, plants were kept under a dark period of 24 hr before images were acquired. Images were acquired 3 days after infiltration (dpi). 35S::GFP:Rab8a and 35S::GFP:ATG8CL lines were produced as described elsewhere (*A Simple and General Method for Transferring Genes into Plants, 1985*) with the pK7WGF2::Rab8a and pK7WGF2::ATG8CL constructs, respectively. Experiments were conducted in *N. benthamiana* leaf epidermal cells unless stated otherwise.

## Pathogenicity assays

*P. infestans* 88069 strain was used in this study. Cultures were grown and maintained by routine passing on rye sucrose agar medium at 18°C in the dark (*van West et al., 1998*). Zoospores were collected from 10 to 14 days old culture by flooding with cold water and incubation at 4°C for 90–120 min. Infection of agroinfiltrated leaves was carried out by addition of 10 mL droplets of zoospore solution at 50,000 spores/ml on detached *N. benthamiana* leaves (*Chaparro-Garcia et al., 2011*). Infection for microscopic experiments carried out on attached leaves. Inoculated detached leaves or plants were kept in humid conditions. Day light/UV images were taken at 7 days post infection and lesion areas were measured in ImageJ.

## Molecular cloning and plasmid constructs

Various constructs used in this study were published previously. GFP:ATG8CL, GFP:PexRD54, GFP:PexRD54aim, RFP:Rem1.3 constructs were previously described in *Bozkurt et al., 2015*. JOKA2: BFP, BFP:EV, BFP:PexD54, BFP:ATG8CL, ATG9:GFP, 3xHA:EV, JOKA2:3xHA constructs were described in *Dagdas et al., 2016*. GFP:ATG8 1.1, GFP:ATG81.2, GFP:ATG8 2.2, GFP:ATG8 3.1, GFP:ATG8 3.2, GFP:ATG8 four were described in *Zess et al., 2019*. RFP:PexRD54, RFP:AIMp, RFP: ATG8CL, RFP:Rab8a, Joka2:RFP, BFP:AIMp, 3xHA:PexRD54, 3xHA:PexRD54aim, and 3xHA: Rd54AIMp constructs were generated by Gibson assembly of each gene PCR fragment into EcoRV digested RFP/tagBFP/HA vectors (N-terminal fusion for PexRD54, PexRD54aim, PexRD54AIMp and ATG8CL, C-terminal fusion for Joka2). For YFP:Oleosin, the eYFP fluorophore was split into N-terminal (residue M1- A155) and C-terminal half (residue D156 - K239). The N-terminal split YFP half was used via a linker peptide RPACKIPNDLKQKVMNH and the C-terminal split YFP half via a linker peptide HNMVKQKLDNPIKCAPR. EcoRV restriction site was added at the end of each linker to allow linearization of the vector and provide an insertion site for subsequent cloning. The DNA fragment encoding Oleosin, together with the linker peptides and restriction sites were amplified from *N. benthamiana* gDNA using primers NYFP-Oleosin F and Oleosin-CYFP R then assembled into pK7WGF2 vector backbone by Gibson assembly. GFP:Rab8a and RFP:Rab8a constructs were generated by PCR amplification from *Solanum tuberosum* cDNA using primers GW_StRab8a-1_F GW_StRab8a-1_R followed by Gateway cloning into the entry vector pENTR/D/TOPO (Invitrogen) then into the pK7WGF2 (GFP) and pH7WGR2 (RFP) vectors, respectively. RFP:GUS was created from the pENTR-GUS control plasmid provided in the GATEWAY cloning kit and inserted into pH7WGR2 (RFP) via LR reaction. Single residue mutations of Rab8aS29N, Rab8aQ74L and Rab8aN128I were obtained by inverse polymerase chain reaction (PCR) amplification of the StRab8a entry clone with the primer pairs (phosphorylated at five prime ends) carrying desired mutations; (i) Rab8aS29N_F and Rab8aS29N_R; (ii) Rab8aQ74L_F and Rab8aQ74L_R; (iii) Rab8aN128I_F and Rab8aN128I_R. Templates were then eliminated by one-hour Dpn-I (New England Biolabs) restriction digestion at 37°C and the PCR products of mutants were ligated using standard protocols to obtain circular Gateway entry clones carrying desired mutations. Next, the entry clones of Rab8a mutants were recombined into destination vectors pK7WGF2 or pB7RWG2 by Gateway LR reaction. All remaining constructs were amplified from existing constructs previously described (*Bozkurt et al., 2015*; *Dagdas et al., 2016*; *Dagdas et al., 2018*), using primer pairs GA_RD54_F with GA_RD54_R for PexRD54, GA_RD54_F with GA_LIR2_R for PexRD54aim, GA_AIMp_F with GA_LIR2_R for PexRD54AIMp, GA_ATG8C_F with GA_ATG8C_R for ATG8CL and GA_NbJoka2_1_Fr with GA_NbJoka2_1_Rv. Silencing constructs for Rab8a were amplified using primer combinations NbRab8A_silF1 and NbRab8A_silR1, Rab8a1-4$^{RNAi}$_F1, Rab8a1-4$^{RNAi}$_F2, Rab8a1-4$^{RNAi}$_F3, Rab8a1-4$^{RNAi}$_R1, Rab8a1-4$^{RNAi}$_R2 and Rab8a1-4$^{RNAi}$_R3, and cloned into the pRNAiGG vector as described elsewhere (*Dagdas et al., 2018*). Silencing of Rab8a was verified using RT-PCR. CFP-GLOX (Glycolate oxidase) was generated by PCR amplification from wheat cDNA using primers GLOX-F and GLOX-R followed by Gateway cloning into the entry vector pENTR/D/TOPO (Invitrogen) then into the pGWB445 (CFP) vector.

## Synthetic construct generation

The sequence for the silencing resistant synthetic Rab8a construct (Rab8a-1$^{syn}$) was obtained by codon shuffling the sequence of Rab8a-1 through a combination of the Integrated DNA

Technologies (IDT) Codon optimization tool and manual codon shuffling. The construct was then gene synthesized and inserted into the vector by Gibson assembly.

## Peptide synthesis

AIMp$^{syn}$ and mAIMp$^{synAA}$ were synthesized with solid phase peptide synthesis and purified with HPLC. 5 (6)-Carboxyfluorescein (Merckmillipore, Chem851082) and 5 (6)-CarboxyTAMRA (Merckmillipore, Chem851030) fluorophores were incorporated with double coupling using DIC/K-Oxyma and HCTU/DIEA.

## Co-immunoprecipitation experiments and immunoblot analysis

Proteins were transiently expressed by agroinfiltration in *N. benthamiana* leaves and harvested 2 days post agroinfiltration. Protein extraction, purification and western blot analysis steps were performed as described previously (*Bozkurt et al., 2011*; *Dagdas et al., 2016*). Polyclonal anti-GFP (Chromotek), anti-tBFP (tRFP) (Evrogen) antibodies produced in rabbit, monoclonal anti-RFP (Chromotek) and anti-FLAG (Sigma-Aldrich) antibodies produced in mouse, monoclonal anti-GFP (Chromotek) and anti-HA (Chromotek) produced in rat were used as primary antibodies. For secondary antibodies anti-mouse antibody (Sigma-Aldrich), anti-rabbit (Sigma-Aldrich) and anti-rat (Sigma-Aldrich) antibodies were used.

## RNA isolation, cDNA synthesis, and RT-PCR

For RNA extraction, 100 mg of leaf tissue was excised and frozen in liquid nitrogen. RNA was then extracted using either GeneJET Plant RNA Purification Kit (Thermo Scientific) or TRIzol RNA Isolation Reagent (Invitrogen) according to producers' recommendations. RNA concentration was measured using NanoDrop Lite Spectrophotometer (Thermo Scientific). For *Figure 2—figure supplement 2 and* µg of extracted RNA underwent DNase treatment using RQ1 RNase-Free DNase (Promega). Two µg of RNA was then used for cDNA synthesis using SuperScript IV Reverse Transcriptase (Invitrogen). cDNA was then amplified using Phusion High-Fidelity DNA Polymerase (New England Biolabs) with the appropriate primer pairs (TKey Resources Table). GAPDH is used to normalize the transcription levels of the genes.

## Confocal microscopy and image processing

Microscopy analyses were carried out on live leaf tissue 3–4 days post agroinfiltration. To minimize the damage of live tissue, leaf discs of *N. benthamiana* were cut using a cork borer and mounted onto Carolina observation gel (Carolina Biological Supply Company). For BODIPY-dodecanoic acid (BODIPY-C12, Invitrogen) staining, 10 µM was infiltrated into the leaf tissue 5 hr prior to observation. For PexRD54 AIM peptide experiments in leaf tissue, a solution of 10 µM of peptide in agroinfiltration buffer or buffer alone was infiltrated in leaves 3 hr prior to observation. For imaging in roots, seedlings were collected at 3 weeks old and the roots placed in 2 mL tubes containing 5 µM peptide solution in agroinfiltration buffer or buffer alone for 3 hr prior to observation. Confocal florescence microscopy was performed using Leica SP5 and SP8 resonant inverted confocal microscope (Leica Microsystems) using 63x and 40x water immersion objective, respectively. In order to excite fluorescent tagged proteins, Diode laser excitation was set to 405 nm, Argon laser to 488 nm and the Helium-Neon laser to 561 nm and their fluorescent emissions detected at 450–480, 495–550, and 570–620 nm to visualize BFP, GFP, and RFP fluorescence, respectively. Sequential scanning between lines was done to avoid spectral mixing from different fluorophores and images acquired using multichannel. Maximum intensity projections of Z-stack images were processed using ImageJ (2.0) to enhance image clarity.

## Data analysis and statistics

Images for quantification of autophagosome numbers were obtained from Z stacks consisting of 1.3 µm depth field multi-layered images with similar settings for all samples unless stated otherwise. To detect and quantify punctate structures in one channel (green channel or red channel or blue channel) and to validate colocalization an overlay of two or three channel, where applicable, was acquired (green channel and/or red channel and/or blue channel). Z stacks were separated into individual images with the ImageJ (2.0) program and analyzed. Autopahgosome quantification was done using

ATG8CL as a marker protein as described before (*Dagdas et al., 2016*). Colocalization of puncta was analyzed using ImageJ (2.0) with a modified version of the colocalization macro described elsewhere (*Pampliega et al., 2013*) with thresholding set as manual to avoid background cytoplasmic signals in each image. Boxplots were generated with mean of punctate numbers generated from stacks obtained in three to six independent biological experiments. Statistical differences were analysed by Welch Two Sample t-test in R. Measurements were significant when p<0.05 (*) and highly significant when p<0.001(***).

## Automated puncta counting algorithm through image processing

The image processing algorithm calculates the gradient of the image to identify the boundaries of the puncta. We then algorithmically identify the enclosed regions formed by the boundaries and counted the number of puncta in each figure. For the case of co-localisation, the co-ordinates of the centres of the puncta/clusters from each channel were calculated and compared to see if they lie within a small tolerance for each puncta and channel. The puncta/clusters satisfying the abovementioned conditions were considered to be co-localized and were counted.

## Acknowledgements

We thank all members of the Bozkurt Lab for their helpful discussion and suggestions. We also are grateful to the Imperial College Facility for Imaging by Light Microscopy (FILM) at Imperial College London for their imaging expertise and technical assistance. This project was funded by the Biotechnology Biological Sciences Research Council (BBSRC) (BB/M002462/1 and BB/M011224/1), the Gatsby Charitable Foundation (GAT3395/GLD), the European Research Council, the Agencia Nacional de Promoción Científica y Tecnológica (ANPCyT, Argentina), the Royal Society (UF160413) and the Austrian Academy of Sciences.

## Additional information

### Funding

| Funder | Grant reference number | Author |
|---|---|---|
| Biotechnology and Biological Sciences Research Council | BB/M002462/1 | Pooja Pandey Tolga O Bozkurt |
| Biotechnology and Biological Sciences Research Council | BB/M011224/1 | Yasin Tumtas Cian Duggan Tolga O Bozkurt |
| Gatsby Charitable Foundation | TSL core grant | Sophien Kamoun |
| Biotechnology and Biological Sciences Research Council | BBS/E/J 000PR9797 | Sophien Kamoun |
| Gatsby Charitable Foundation | GAT3395/GLD | Temur Yunusov Sebastian Schornack |
| Royal Society | UF110073 UF160413 | Sebastian Schornack |
| Austrian Academy of Sciences | SFB F79 | Yasin Dagdas |
| H2020 European Research Council | ANPCyT | Sophien Kamoun |

The funders had no role in study design, data collection and interpretation, or the decision to submit the work for publication.

### Author contributions

Pooja Pandey, Data curation, Software, Formal analysis, Validation, Investigation, Visualization, Methodology, Writing - original draft, Writing - review and editing; Alexandre Y Leary, Conceptualization, Resources, Data curation, Software, Formal analysis, Supervision, Validation, Investigation, Visualization, Methodology, Writing - original draft, Writing - review and editing; Yasin Tumtas, Conceptualization, Resources, Data curation, Software, Formal analysis, Validation, Investigation,

Visualization, Methodology, Writing - original draft, Writing - review and editing; Zachary Savage, Conceptualization, Resources, Formal analysis, Investigation, Methodology, Writing - original draft, Writing - review and editing; Bayantes Dagvadorj, Formal analysis, Investigation, Visualization, Methodology; Cian Duggan, Resources, Formal analysis, Investigation, Visualization, Methodology, Writing - original draft, Writing - review and editing; Enoch LH Yuen, Resources, Data curation, Formal analysis, Investigation, Methodology, Writing - original draft, Writing - review and editing; Nattapong Sanguankiattichai, Federico Gabriel Mirkin, Conceptualization, Resources, Investigation, Methodology, Writing - review and editing; Emily Tan, Amber J Connerton, Data curation, Formal analysis, Investigation, Methodology; Virendrasinh Khandare, Resources, Formal analysis, Investigation, Methodology; Temur Yunusov, Mathias Madalinski, Resources, Investigation, Methodology; Sebastian Schornack, Resources, Supervision, Writing - original draft, Writing - review and editing; Yasin Dagdas, Sophien Kamoun, Conceptualization, Resources, Supervision, Writing - original draft, Writing - review and editing; Tolga O Bozkurt, Conceptualization, Resources, Data curation, Software, Formal analysis, Supervision, Funding acquisition, Validation, Investigation, Visualization, Methodology, Writing - original draft, Project administration, Writing - review and editing

### Author ORCIDs

Pooja Pandey (iD) https://orcid.org/0000-0003-3145-7794
Alexandre Y Leary (iD) http://orcid.org/0000-0001-7223-3557
Bayantes Dagvadorj (iD) http://orcid.org/0000-0002-0188-9353
Cian Duggan (iD) http://orcid.org/0000-0001-7302-7472
Enoch LH Yuen (iD) https://orcid.org/0000-0002-7933-0605
Virendrasinh Khandare (iD) http://orcid.org/0000-0003-2673-6561
Sebastian Schornack (iD) http://orcid.org/0000-0002-7836-5881
Yasin Dagdas (iD) http://orcid.org/0000-0002-9502-355X
Sophien Kamoun (iD) http://orcid.org/0000-0002-0290-0315
Tolga O Bozkurt (iD) https://orcid.org/0000-0003-0507-6875

### Decision letter and Author response

Decision letter https://doi.org/10.7554/eLife.65285.sa1
Author response https://doi.org/10.7554/eLife.65285.sa2

## Additional files

### Supplementary files

- Source data 1. Statistical_Summary for all figures.

- Transparent reporting form

### Data availability

All data generated or analysed during this study are included in the manuscript and supporting files. Source data files have been provided for Figures 1-9.

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
