## [Decision Letter]

**Acceptance summary:**

This study provides mechanistic insights on how the oomycete pathogen *Phytophthora infestans* exploits host plant autophagy for its own benefit. The authors show that the *P. infestans* effector PexRD54 links the host vesicle transport regulator Rab8a to autophagosome biogenesis at the pathogen interface, suggesting Rab8a to be a new autophagy component. By using the effector as a probe, the authors identify a novel genetically encoded tool (AIMp) to study the role of autophagy, which functions as an autophagy inhibitor. This will have a major impact on autophagy research in plants (and maybe beyond if the AIMp works across kingdoms).

**Decision letter after peer review:**

Thank you for submitting your article "An oomycete effector subverts host vesicle trafficking to channel starvation-induced autophagy to the pathogen interface" for consideration by *eLife*. Your article has been reviewed by 3 peer reviewers, and the evaluation has been overseen by a Reviewing Editor and Jürgen Kleine-Vehn as the Senior Editor. The following individuals involved in review of your submission have agreed to reveal their identity: Takashi Ueda (Reviewer #2); Elena A. Minina (Reviewer #3).

All reviewers are enthusiastic about your manuscript, but some parts still need to be improved.

Here you find a list of essential revisions for the manuscript to become acceptable. I am also attaching the full list of reviewers comments for your reference.

When preparing your resubmission please:

1) Clarify how AIMp connects to Rab8.

2) Provide evidence that PexRD54 interacts with Nicotiana Rab8.

3) Provide additional experimental evidence that PexRD54 triggers an autophagy response similar to carbon starvation, or tone down the statements related to this aspect.

4) Determine whether RAB8 autophagosomes induced by PexRD54 or infection.

5) Evaluate a possible dominant-negative effect of RAB8aNI and whether they affect lesion formation by the pathogen.

6) Elaborate/discuss more on how PexRD54 may boost pathogenicity although it is degraded in the vacuole and try to accommodate reviewer #3's comments regarding the discussion.

*Reviewer #1:*

In the present manuscript Pandey, Leary et al. elucidate how Phytophthora effector PexRD54 is activating autophagy to increase pathogenicity of Phytophthora in plants. Previous research from the group of Tolga Bozkurt already identified how PexRD54 counteracts Joka2-dependent selective autophagy and hence anti-microbial autophagy. However, the exact mechanism of how this effector activates autophagy and additional pro-microbial functions of autophagy during Phytophthora-host interaction remain elusive. The authors first clarify that the AIM motif of PexRD54 is not the feature that is responsible for autophagy activation. Surprisingly, they identify that this motif inactivates autophagy (and thus restricts pathogenicity of Phytophthora), which will be a valuable tool for the autophagy community in the future. Previous interaction studies revealed that a small Ras-related GTPase, Rab8 might interact with RexRD54 as it interacts with ATG8CL. Indeed, Rab8 associates with RexRD54 independent of its capacity to bind ATG8. Its association with autophagosomes is further increased by PexRD54. Loss of function mutants of Rab8 result in an impaired PexRD54-dependent induction of autophagosomes and provide a first evidence that Rab8 is required for PexRD54-triggered autophagy. The effector seems to activate autophagy that resembles carbon-starvation induced autophagy, which seems to be dependent on Rab8.

The paper is very well-written and provides a detailed, mechanistic analysis of the function of PexRD54. It proves, how powerful it is to work with effectors as tools to identify novel (autophagy) components. The conclusions of this paper are well supported by data, but some aspects need clarification or small additional control experiments.

Comments:

1) By using effectors as probes, the authors identify a novel tool (AIMp) to study the role of autophagy and also reveal a novel autophagy component. Both will have a major impact on the autophagy community in plants (and maybe beyond if the AIMp works across kingdoms). However, to me it is not 100% clear how the first part about the AIMp really connects to Rab8 and also subsequent analysis of carbon starvation-triggered autophagy. The authors claim at the end of the section that their analysis of the AIMp reveals that there must be other features within PexRD54 to stimulate autophagy (like association with Rab8). But AIMs are not really the autophagy-activating motif, they mediated interaction with ATG8 and enable specific degradation of cargos. Also, little evidence is provided how AIMp functions to block autophagy (additional control experiments might reveal this) and if its effect is really broad (other plant species) or only limited to *N. benthamiana*.

2) With Rab8, a small ras-related GTPase, the authors identify a novel autophagy component that might be involved in autophagosome biogenesis. The analysis of Rab8 is detailed and provides mechanistic insights how Rab8 contributes to the function of PexRD54 to boost autophagy. Most of the analysis is done with the potato Rab8. The in-planta analysis of Rab8 function in autophagy is performed by silencing Rab8 in *N. benthamiana*. However, no evidence is provided whether PexRD54 is interacting with the four NbRab8 proteins or whether Rab8 from potato and tobacco are very similar (phylogenetic analysis or similar).

3) In addition, to claim that RexRD54 is triggering an autophagy-response similar to carbon-starvation induced autophagy is a bit overinterpreted and lacks further experimental evidence. Additional proof is missing that the short dark period really induces carbon starvation in *N. benthamiana* leaves. PexRD54 might activate multiple (selective) autophagy pathways that are beneficial for the pathogen as autophagy has a very broad role and is implicated in many cellular processes. Additional small experiments (further analysis of hallmarks of carbon starvation e.g., induction of chlorophagy etc.) or adding a more in-depth discussion about the possibility that PexRD54 might induce more than carbon-starvation-triggered autophagy, will improve this section.

Specific points to the authors:

1) The section about the AIMp would profit from some additional small control experiments:

-Gene expression analysis of ATG8 and Joka2 (to exclude it is due to transcriptional activation of both genes).

-Block of autophagy in autophagy-inducing conditions: e.g. carbon starvation or AZD treatment in plants. Is it possible it blocks only specific autophagy responses?

2) Does the AIMp also work in Arabidopsis? I guess this would be great to know for the autophagy community and would highlight the broad function of the AIMp. It would be also good to elaborate a bit on the potential mechanism how AIMp inactivates autophagy. As autophagosomes are less formed it looks like biogenesis is affected, did the authors already investigate this or is this subject of future research?

3) The authors provide evidence that PexRD54 ends up in the vacuole and is degraded eventually. How does this fit into its function to boost pathogenicity and diverting autophagy to the infection interface to provide the pathogen with "nutrients"? If the effector is degraded it might not be able to boost autophagy anymore. I would elaborate on this a bit more.

4) Silencing of Rab8 in *N. benthamiana* nicely demonstrate that it is required for PexRD54-triggered autophagy. How similar are the Rab8s to the one from potato that was used in the interaction studies? Do they interact with ATG8 and with PexRD54? Considering also that there are 4 and silencing of all of them but not of 1+2 influences basal autophagy, it would be interesting to see whether PexRD54 interacts only with a subset of NbRab8s. I also don't agree with the last sentence on p15. As I understand silencing of 1+2 is not impacting basal autophagy but silencing of all of them does. Did the authors target 3+4 only to elucidate this? They might not have redundant function in the end if 2 influence basal autophagy and the other ones not.

5) In the last section, the authors conclude that PexRD54 triggers an autophagy response similar to carbon-starvation induced autophagy. McLoughlin et al., 2020 (TPC) covered maize leaves for 2d to induced carbon starvation. Of course, maize and *N. benthamiana* are different organisms but is 24 hours really sufficient to trigger a carbon starvation response? No evidence, apart from ATG8 puncta accumulation (which is also quite mild) is provided that carbon starvation is really ongoing. Theoretically, autophagy inducing conditions degrade ATG8 also more rapidly, thus inhibition of vacuolar degradation (e.g., E64d, or ConcA) should be included to assess the real flux of ATG8 during autophagy inducing conditions. It is also possible to provide transcriptional upregulation of the autophagy pathway during these conditions. The authors provide support for their carbon-starvation theory by showing that BODIPY co-localizes stronger with Rab8 in the presence of PexRD54. Still, I think that more data is required to be able to conclude that PexRD54 is triggering a carbon-starvation-like response (looking into chlorophagy, amino acid accumulation, sugars, metabolome analysis). I guess that this is not really the scope of this manuscript. Alternatively, the authors could add control experiments with AZD8055 or Tm (e.g., analyzing the BODIPY co-localization) and/or discuss the possibility that other autophagy responses might be triggered by PexRD54 that are beneficial for the pathogen.

*Reviewer #2:*

In this manuscript, authors analyzed biological significance of the interaction between an effector protein PexRD54 secreted by the Irish famine pathogen *Phytophthora infestans* and a host membrane trafficking regulator RAB8a. Authors demonstrated that expression of the ATG8 interacting motif (AIM) of PexRD54 and full-length PexRD54 distinctly affected autophagosome formation in N. bethamiana, and speculated that the distinct effects should be conferred by interaction of full-length PexRD54 with additional factors to ATG8. Authors then showed that the GDP-bound state of RAB8a, a member of RAB GTPases acting in membrane trafficking, interacted with PexRD54 at a distinct region from the AIM, and showed that RAB8a was required for PexRD54-induced autophagy, whereas RAB8a was not involved in Joka2-mediated antimicrobial autophagy. Autophagosomes induced by PexRD54 were stained by Bodipy C-12, and lipid droplets were observed closely associated to the autophagosomes. RAB8a knock down resulted in reduced association between PexRD54-positive autophagosomes and LDs, suggesting that RAB8a promoted recruitment of LDs to autophagosome formation sites. Furthermore, authors' finding that the AIM peptide from PexRD54 acts as a general inhibitor of autophagy would provide a powerful methodology to modulate autophagic activity at any tissues and developmental stages of users' interest.

The reported mechanism of autophagosome formation involving RAB8 and LDs induced by the pathogen is novel and interesting. A few supporting data addressing questions listed below, mainly on the RAB8 function, could make this work more convincing and solid.

a. Regarding the RAB8-positive punctate compartments. RAB8 has been localized to the Golgi apparatus (Speth et al., 2009, Plant Phys), peroxisome (Cui et al., 2013, JBC), and SCD-positive compartments (Meyers et al., 2017, Plant Cell). Are these compartments with RAB8 are recruited to autophagosome formation sites induced by PexRD54 or infection? Otherwise, RAB8a is recruited there irrespective of these compartments?

b. As long as I know, RAB GTPases with the N-to-I mutation frequently act as a dominant negative form because this mutant titrates molecules that preferentially interact with GDP-bound and/or nucleotide-free forms of RAB GTPases such as GEFs, similar to the GDP-freeze S-to-N mutant. Could authors provide discussion regarding why these mutant versions of RAB8a conferred distinct effects on autophagosome formation induced by PexRD54?

c. Related to the above, could authors include the result of the ATG8CL puncta in cells expressing HA:PexRD54 without GFP-RAB8a co-expression, which would be needed to evaluate the dominant-negative effect of the RAB8aNI mutant?

d. For the experiment presented in Figure 8—figure supplement 3. Have authors observed other organelles in this experiment? The data of organelles, which do not accumulate with RAB8a and LDs, would be a nice negative control.

e. I expect that mutant versions of RAB8a would also affect lesion formation when co-injected with the pathogen. Could authors provide a data using RAB8a such that presented in Figure 9C?

*Reviewer #3:*

In this study authors provide a mechanistic insight on the interplay between a *Phytophthora infestans* effector protein PexRD54, plant endomembrane trafficking system and the major catabolic pathway – autophagy. Authors build on their previous observations to demonstrate how the effector protein reroutes trafficking of host endomembranes to the benefit of the pathogen. Using a combination of mutagenesis, biochemistry and advanced microscopy methods authors show that PexRD54 effector protein triggers recruitment of plant Rab8 small GTPase and lipid droplets for formation of autophagosomes, which are then diverted from the typical route to degradation and are potentially delivering nutrients to the pathogen. Furthermore, authors confirm the physiological relevance of the molecular biology results for the *P. infestans* infection efficacy. These study adds a valuable insight on the complex role of the plant autophagy in plant pathogen interactions and demonstrates a potential across-kingdoms conservation of the Rab8 role in regulating fusion of lipid droplets and in polarized re-distribution of endomembranes.

The data is generally of a high quality and authors provide thorough verification for each hypothesis by implementing combinations of cell biology and biochemistry assays. To further strengthen the study authors could address the following issues:

1. Co-localization analyses could be more comprehensively conducted by implementing standard object- or pixel-based high throughput protocols.

2. Authors conclude that PexRD54 imitates carbon starvation conditions that trigger a specific type of autophagic response. However, no other types of autophagy-inducing conditions were compared to PexRD54 mode of actions.

3. PexRD54 is not only diverting autophagy but is also being degraded by it. To better clarify the dynamics of this balance, it would be beneficial to add to the current data a comparison of differences in the rate of autophagosome delivery to the vacuole under carbon starvation-induced autophagy and autophagy induced by PexRD54.

The manuscript is very well structured and written, the logic of the results is easy to follow and majority of the data is of a high quality.

---

## [Author Response]

All reviewers are enthusiastic about your manuscript, but some parts still need to be improved.Here you find a list of essential revisions for the manuscript to become acceptable. I am also attaching the full list of reviewers comments for your reference.When preparing your resubmission please:1) Clarify how AIMp connects to Rab8.

We updated the discussion to explain this as follows:

“Here, the AIMp served as an excellent negative control to understand PexRd54 activities and to interpret the impact of autophagy manipulation by the pathogen. We used it as a tool to perturb PexRD54 activities without directly interfering with Rab8a functions. Most ATG mutants show pleiotropic affects and have non-autophagy related roles, making it difficult to interpret the outcomes of autophagy inhibition. However, the AIMp is a great tool to block autophagy specifically, as it directly acts on ATG8.”

2) Provide evidence that PexRD54 interacts with Nicotiana Rab8.

We provide new data that shows PexRD54-NbRab8a interaction (Figure 3—figure supplement 1)

3) Provide additional experimental evidence that PexRD54 triggers an autophagy response similar to carbon starvation, or tone down the statements related to this aspect.

In the manuscript, we did not only compare PexRD54 triggered responses to carbon starvation. We also tested autophagy activation by the aggrephagy cargo receptor Joka2 (Figure 6-7). We also had the data that there is no additive effects of carbon starvation or PexRD54 on ATG8 puncta (Figure 7A-B). Based on these findings, we concluded that PexRD54 triggered autophagy is reminiscent of autophagy induced by carbon starvation, but not to autophagy induced by the aggrephagy receptor Joka2/NBR1 (Jung et al., 2019, Journal of Experimental Botany). We revised the text to better explain this point in the relevant results and Discussion sections

In the earlier version of the manuscript, we quantified ATG8 puncta to determine the effect of carbon starvation. We now include new data that 24 hours of light restriction leads to enhanced ATG8 depletion (Figure 7—figure supplement 1)

However, we also agree that we have not tested whether PexRD54 could trigger responses similar to other potential autophagy inducing conditions. Therefore, we revised the text accordingly and toned down the statements as follows:

“Intriguingly, the autophagy pathway primed by PexRD54 resembles autophagy induced by carbon starvation, but not to autophagy induced by the aggrephagy cargo receptor Joka2. In contrast to Joka2, both PexRD54 and light restriction stimulated enhanced Rab8a-ATG8CL and Rab8a-LD associations (Figure 6 and Figure 7). Furthermore, their effects on autophagosome formation were not additive (Figure 7A-B)…… Therefore, we cannot exclude the possibility that PexRD54 could activate other host autophagy pathways that are beneficial to the pathogen. Further research is needed to determine whether PexRD54 could mimic other autophagy inducing conditions and exploit other host autophagy pathways.”

“Nevertheless, further assays are needed to determine whether Rab8a is implicated in other autophagy pathways.”

“However, it is possible that PexRD54 could activate other host autophagy pathways that favor pathogen virulence.”

4) Determine whether RAB8 autophagosomes induced by PexRD54 or infection.

We detected at least two different sets of Rab8a puncta around the haustorium; some that are labeled by ATG8CL and some that are not (Figure 8A-D, Figure 8—figure supplement 2). In haustoriated cells AIMp only blocked the Rab8a puncta that is positive for ATG8CL (Figure 8B, Figure 8—figure supplement 2), indicating that Rab8a autophagosomes are induced by PexRD54. Likewise, upon overexpressing Joka2, which competes with PexRD54 for ATG8CL binding (Figure 8D), we did not see any ATG8CL puncta that is positive for Rab8a around the haustorium. Our data that PexRD54 expression (in the absence of infection) boosts Rab8a-ATG8CL colocalization and association (Figure 4) further supports the notion that Rab8 autophagosomes are induced by PexRD54.

In addition, Rev2 highlighted that other Rab8 members (RabE1 family) associate with Golgi and peroxisomes in Arabidopsis, and asked whether Rab8a autophagosome recruitment in infected cells could be associated with these compartments. As suggested, we checked the potential colocalization of perihaustorial Rab8a with organelle markers. We now provide data that Rab8a localizes to Golgi but not to peroxisomes or to mitochondria in both infected and uninfected cells. However, perihaustorial Rab8a/PexRD54 autophagosomes are not labeled by the Golgi marker (Figure 8—figure supplement 3). This also confirms our finding that at least two distinct Rab8a puncta target the pathogen interface and PexRD54 possibly directs a subpopulation of Rab8a from Golgi to PexRD54/ATG8CL autophagosomes. Therefore, we conclude that Rab8a localization to PexRD54 autophagosomes cannot be explained by Rab8a’s association with these organelles.

5) Evaluate a possible dominant-negative effect of RAB8aNI and whether they affect lesion formation by the pathogen.

We provide data that RAB8aNI has a dominant negative affect on PexRD54 triggered autophagy (Figure 5—figure supplement 8).

We also provide data that RAB8aNI leads to more susceptibility. Consistent with this data, silencing of Rab8a1-4 leads to enhanced susceptibly. We were able to rescue this phenotype by overexpressing a silencing resilient synthetic construct of Rab8a (Figure 8—figure supplements 6-8) These findings suggest that Rab8a plays a positive role in immunity, whereas PexRD54 could interfere with this process by diverting a subpopulation of Rab8a to the autophagosomes.

We updated the text accordingly and included a new section about the “role of Rab8a in immunity” in the discussion.

6) Elaborate/discuss more on how PexRD54 may boost pathogenicity although it is degraded in the vacuole and try to accommodate reviewer #3's comments regarding the discussion.

We explained this by expanding the Discussion section (Page 34, “why does *P. infestans activate autophagy?)” as follows:*

“… However, our data suggests that the fate of PexRD54 autophagosomes in infected and uninfected cells could be different, since in uninfected cells PexRD54 is degraded inside the plant vacuole following 4-6 days of ectopic expression (Figure 2—figure supplement 1-3). Therefore, PexRD54 may or may not be degraded in infected plant cells during infection. PexRD54 is a member of RXLR effectors which typically show high level of expression during 2-5 days of infection. Consequently, even if PexRD54 is depleted via host autophagy machinery during infection, there would be more of it secreted via the haustorium to maintain PexRD54 virulence functions as long as the haustorium is accommodated in the host cells. PexRD54 likely remodels the cargoes engulfed in ATG8CL^-^autophagosomes targeted to pathogen interface, which are subsequently assimilated by the parasite. Whether these cargoes are hydrolyzed in infected plant cells or directly absorbed by the pathogen remains to be determined.”

Reviewer #1:Comments:1) By using effectors as probes, the authors identify a novel tool (AIMp) to study the role of autophagy and also reveal a novel autophagy component. Both will have a major impact on the autophagy community in plants (and maybe beyond if the AIMp works across kingdoms). However, to me it is not 100% clear how the first part about the AIMp really connects to Rab8 and also subsequent analysis of carbon starvation-triggered autophagy. The authors claim at the end of the section that their analysis of the AIMp reveals that there must be other features within PexRD54 to stimulate autophagy (like association with Rab8). But AIMs are not really the autophagy-activating motif, they mediated interaction with ATG8 and enable specific degradation of cargos. Also, little evidence is provided how AIMp functions to block autophagy (additional control experiments might reveal this) and if its effect is really broad (other plant species) or only limited to N. benthamiana.

We thank the reviewer for the positive evaluation. We expanded the discussion to explain How AIMp connects Rab8a and autophagy (Page 32). Please see responses to the editor point 1.

We agree with the reviewer that AIMs are not necessarily autophagy activating motifs. Our data that AIMp blocks autophagy activation is consistent with this notion. However, we already showed in an earlier study (Dagdas et al., 2016) that PexRD54 and Joka2 AIM motifs are required to stimulate ATG8CL puncta formation. This indicates that these proteins have additional features other than having an AIMp to stimulate autophagosome formation. This is also consistent with the recent findings by Chang *et al.* (Chang et al., 2021, Science Advances) which showed that autophagy cargo receptors, which bind ATG8 via AIMs, can stimulate ATG8 lipidation via different mechanisms.

We believe that the AIMp blocks autophagy by occupying the AIM binding pocket on ATG8 and thereby competing out the autophagy adaptors/receptors required to stimulate autophagy. We now provide new data that AIMp mutant (mutated in the conserved AIM motif -WEIV to AEIA) is unable to prevent ATG8 depletion (Figure 2—figure supplement 4).

Indeed, we did not provide any data that whether AIMp can block autophagy in other systems. Our collaborator Yasin Dagdas’ lab has preliminary data that AIMp works in Arabidopsis, which will be addressed in follow up manuscript.

2) With Rab8, a small ras-related GTPase, the authors identify a novel autophagy component that might be involved in autophagosome biogenesis. The analysis of Rab8 is detailed and provides mechanistic insights how Rab8 contributes to the function of PexRD54 to boost autophagy. Most of the analysis is done with the potato Rab8. The in-planta analysis of Rab8 function in autophagy is performed by silencing Rab8 in N. benthamiana. However, no evidence is provided whether PexRD54 is interacting with the four NbRab8 proteins or whether Rab8 from potato and tobacco are very similar (phylogenetic analysis or similar).

We provide data that PexRD54 interacts with NbRab8a-1 protein (Figure 3—figure supplement 1)

3) In addition, to claim that RexRD54 is triggering an autophagy-response similar to carbon-starvation induced autophagy is a bit overinterpreted and lacks further experimental evidence. Additional proof is missing that the short dark period really induces carbon starvation in N. benthamiana leaves. PexRD54 might activate multiple (selective) autophagy pathways that are beneficial for the pathogen as autophagy has a very broad role and is implicated in many cellular processes. Additional small experiments (further analysis of hallmarks of carbon starvation e.g., induction of chlorophagy etc.) or adding a more in-depth discussion about the possibility that PexRD54 might induce more than carbon-starvation-triggered autophagy, will improve this section.

We updated the text accordingly and included these possibilities (please see responses to editor’s request in point 3). We originally quantified ATG8 puncta to measure the effect of carbon starvation (Figure 7A-B). We also provide new data that ATG8 depletion is enhanced by light restriction (Figure 7—figure supplement 1).

Specific points to the authors:1) The section about the AIMp would profit from some additional small control experiments:-Gene expression analysis of ATG8 and Joka2 (to exclude it is due to transcriptional activation of both genes).

This is unlikely because we see the same effect on ATG8 (Figure 2) and Joka2 when they are expressed using constitutive 35S promoter. We provide new data 35S::Joka2-HA stability is enhanced by the AIMp (Figure 2—figure supplement 5)

-Block of autophagy in autophagy-inducing conditions: e.g. carbon starvation or AZD treatment in plants. Is it possible it blocks only specific autophagy responses?

The AIMp blocks both basal autophagy and Joka2-triggered autophagy (aggrephagy). Since it directly binds and acts on the ATG8, AIMp should be able to block any autophagy pathway that requires ATG8. We also provide new data that AIMp mutant does not stabilize ATG8 (Figure 2—figure supplement 4), providing further support for this notion.

2) Does the AIMp also work in Arabidopsis? I guess this would be great to know for the autophagy community and would highlight the broad function of the AIMp. It would be also good to elaborate a bit on the potential mechanism how AIMp inactivates autophagy. As autophagosomes are less formed it looks like biogenesis is affected, did the authors already investigate this or is this subject of future research?

Our collaborator’s lab (Yasin Dagdas) produced preliminary data that it works in Arabidopsis as well. This will be reported in a future publication. We added additional text in the discussion to elaborate the mechanism of AIMp to block autophagy (Page 32).

3) The authors provide evidence that PexRD54 ends up in the vacuole and is degraded eventually. How does this fit into its function to boost pathogenicity and diverting autophagy to the infection interface to provide the pathogen with "nutrients"? If the effector is degraded it might not be able to boost autophagy anymore. I would elaborate on this a bit more.

We revised the text to explain this. Please see the responses to editor’s comments (point 6)

4) Silencing of Rab8 in N. benthamiana nicely demonstrate that it is required for PexRD54-triggered autophagy. How similar are the Rab8s to the one from potato that was used in the interaction studies? Do they interact with ATG8 and with PexRD54? Considering also that there are 4 and silencing of all of them but not of 1+2 influences basal autophagy, it would be interesting to see whether PexRD54 interacts only with a subset of NbRab8s. I also don't agree with the last sentence on p15. As I understand silencing of 1+2 is not impacting basal autophagy but silencing of all of them does. Did the authors target 3+4 only to elucidate this? They might not have redundant function in the end if 2 influence basal autophagy and the other ones not.

We agree, this is a very good point. As we have not performed silencing of 3+4, we cannot exclude the possibility that Rab8-3+4 contribute to basal autophagy but not Rab8-1-2. We are planning to investigate the potential roles of Rab8a members in basal autophagy in the future. We revised the text accordingly as follows “This suggests a potential redundancy in Rab8a function in basal autophagy. Alternatively, NbRab8a 3-4 might be involved in basal autophagy whereas NbRab1-2 are not, which needs to be tested in future.”

5) In the last section, the authors conclude that PexRD54 triggers an autophagy response similar to carbon-starvation induced autophagy. McLoughlin et al., 2020 (TPC) covered maize leaves for 2d to induced carbon starvation. Of course, maize and N. benthamiana are different organisms but is 24 hours really sufficient to trigger a carbon starvation response? No evidence, apart from ATG8 puncta accumulation (which is also quite mild) is provided that carbon starvation is really ongoing. Theoretically, autophagy inducing conditions degrade ATG8 also more rapidly, thus inhibition of vacuolar degradation (e.g., E64d, or ConcA) should be included to assess the real flux of ATG8 during autophagy inducing conditions. It is also possible to provide transcriptional upregulation of the autophagy pathway during these conditions. The authors provide support for their carbon-starvation theory by showing that BODIPY co-localizes stronger with Rab8 in the presence of PexRD54. Still, I think that more data is required to be able to conclude that PexRD54 is triggering a carbon-starvation-like response (looking into chlorophagy, amino acid accumulation, sugars, metabolome analysis). I guess that this is not really the scope of this manuscript. Alternatively, the authors could add control experiments with AZD8055 or Tm (e.g., analyzing the BODIPY co-localization) and/or discuss the possibility that other autophagy responses might be triggered by PexRD54 that are beneficial for the pathogen.

Overall, we agree with these points and we revised the text accordingly as explained in response to editor’s point no 3. We now provide evidence that ATG8 depletion is enhanced by light restriction (Figure 7—figure supplement 1). Throughout this work, we avoided using E64d and ConcA, as we did not obtain consistent results with these inhibitors. It is likely that they may not work with the same efficiency in *N. benthamiana* compared to Arabidopsis. There is also potential interference by Agrobacterium.

Reviewer #2:In this manuscript, authors analyzed biological significance of the interaction between an effector protein PexRD54 secreted by the Irish famine pathogen Phytophthora infestans and a host membrane trafficking regulator RAB8a. Authors demonstrated that expression of the ATG8 interacting motif (AIM) of PexRD54 and full-length PexRD54 distinctly affected autophagosome formation in N. bethamiana, and speculated that the distinct effects should be conferred by interaction of full-length PexRD54 with additional factors to ATG8. Authors then showed that the GDP-bound state of RAB8a, a member of RAB GTPases acting in membrane trafficking, interacted with PexRD54 at a distinct region from the AIM, and showed that RAB8a was required for PexRD54-induced autophagy, whereas RAB8a was not involved in Joka2-mediated antimicrobial autophagy. Autophagosomes induced by PexRD54 were stained by Bodipy C-12, and lipid droplets were observed closely associated to the autophagosomes. RAB8a knock down resulted in reduced association between PexRD54-positive autophagosomes and LDs, suggesting that RAB8a promoted recruitment of LDs to autophagosome formation sites. Furthermore, authors' finding that the AIM peptide from PexRD54 acts as a general inhibitor of autophagy would provide a powerful methodology to modulate autophagic activity at any tissues and developmental stages of users' interest.The reported mechanism of autophagosome formation involving RAB8 and LDs induced by the pathogen is novel and interesting. A few supporting data addressing questions listed below, mainly on the RAB8 function, could make this work more convincing and solid.a. Regarding the RAB8-positive punctate compartments. RAB8 has been localized to the Golgi apparatus (Speth et al., 2009, Plant Phys), peroxisome (Cui et al., 2013, JBC), and SCD-positive compartments (Meyers et al., 2017, Plant Cell). Are these compartments with RAB8 are recruited to autophagosome formation sites induced by PexRD54 or infection? Otherwise, RAB8a is recruited there irrespective of these compartments?

We thank the reviewer for their careful evaluation and constructive criticism. As suggested, we checked the potential colocalization of perihaustorial Rab8a with organelle markers. We now provide data that Rab8a localizes to Golgi but not to peroxisomes or to mitochondria in both infected and uninfected cells. However, perihaustorial Rab8a/PexRD54 autophagosomes are not labeled by the Golgi marker (Figure 8—figure supplement 3). This also confirms our finding that at least two distinct Rab8a puncta target the pathogen interface and PexRD54 possibly directs a subpopulation of Rab8a from Golgi to PexRD54/ATG8CL autophagosomes. Therefore, we conclude that Rab8a localization to PexRD54 autophagosomes cannot be explained by Rab8a’s association with these organelles. (The results with the SCD marker were inconclusive as the construct for this we obtained was using the native Arabidopsis promoter and did not produce any signal in *N. benthamiana* in our hands)

In haustoriated cells AIMp only blocked the Rab8a puncta that is positive for ATG8CL (Figure 8B, Figure 8—figure supplement 2), indicating that Rab8a autophagosomes are induced by PexRD54. Likewise, upon overexpression of Joka2, which competes with PexRD54 for ATG8CL binding (Figure 8D), we did not see any ATG8CL puncta that is positive for Rab8a around the haustorium. Our data that PexRD54 expression (in the absence of infection) boosts Rab8a-ATG8CL colocalization and association (Figure 4) further supports the notion that Rab8 autophagosomes are induced by PexRD54.

b. As long as I know, RAB GTPases with the N-to-I mutation frequently act as a dominant negative form because this mutant titrates molecules that preferentially interact with GDP-bound and/or nucleotide-free forms of RAB GTPases such as GEFs, similar to the GDP-freeze S-to-N mutant. Could authors provide discussion regarding why these mutant versions of RAB8a conferred distinct effects on autophagosome formation induced by PexRD54?

As highlighted by the reviewer Rab GTPase N-to-I mutants generally act as dominant negative forms and our data with this mutant is consistent with this notion (Figure 5—figure supplement 7-8). We were initially surprised by our data that S29N mutant showed enhanced PexRD54 puncta whereas the Q74L mutant did not. However, earlier work showed that these mutations cannot be generalized and could have different outcomes when introduced into different Rabs (Langemeyer et al., 2014, *eLife*; Pfeffer et al., 2014, e*Life*). Consitent with this view, Rab8-GDP mutant was found to have an unusual acitivty to promote LD fusions in mamallian cells, indicating that S-to-N mutation in this Rab functions differently. We included additional information in the relevant result section as follows:

“These findings were initially surprising given the general view that Rab S-to-N mutations in this position lead to less active (or inactive) forms that mimic the GDP bound state whereas the Q-to-L mutations in this site are assumed to be locked in GTP bound state that is more active. However, there are reports which revealed that these mutations cannot be generalized (Langemeyer et al., 2014; Pfeffer et al., 2014). Consistent with this view, the GDP bound form of the mammalian Rab8a was found to promote lipid droplet (LD) fusions, indicating that S-to-N mutation in this Rab functions differently. Therefore, further biochemical evidence is required to determine whether these mutations show perturbed GTPase activities.”

c. Related to the above, could authors include the result of the ATG8CL puncta in cells expressing HA:PexRD54 without GFP-RAB8a co-expression, which would be needed to evaluate the dominant-negative effect of the RAB8aNI mutant?

We agree this would be required to determine the dominant negative effect. We now provide this data in (Figure 5—figure supplement 8)

d. For the experiment presented in Figure 8—figure supplement 3. Have authors observed other organelles in this experiment? The data of organelles, which do not accumulate with RAB8a and LDs, would be a nice negative control.

This is a good suggestion. We now included new panel in which mitochondria marker is used as a negative control. Please note that this is now labeled as Figure 8—figure supplement 4.

e. I expect that mutant versions of RAB8a would also affect lesion formation when co-injected with the pathogen. Could authors provide a data using RAB8a such that presented in Figure 9C?

We provide data that RAB8aNI leads to more susceptibility. Consistent with this data, silencing of Rab8a1-4 leads to enhanced susceptibly. We were able to rescue this phenotype by overexpressing a silencing resilient synthetic construct of Rab8a (Figure 8—figure supplements 6-8) These findings suggest that Rab8a plays a positive role in immunity, whereas PexRD54 could interfere with this process by diverting a subpopulation of Rab8a to the autophagosomes.

We updated the text accordingly and included a new section about the “role of Rab8a in immunity” in the discussion.

Reviewer #3:The data is generally of a high quality and authors provide thorough verification for each hypothesis by implementing combinations of cell biology and biochemistry assays. To further strengthen the study authors could address the following issues:1. Co-localization analyses could be more comprehensively conducted by implementing standard object- or pixel-based high throughput protocols.

We agree that pixel-based colocalization analysis would generally be ideal to determine precise localization of fluorescently labeled proteins. In our manuscript, we used the colocalization macro reported in (Pampliega *et al.,* 2013, Nature) that is ideal to measure red/green puncta to quantify autophagosome colocalization. We were mainly interested in colocalization at the punctate structures, however the proteins that we study (PexRD4, Rab8a, ATG8CL, Joka2) have background cytoplasmic signals which perturbed colocalization analysis when using pixel-based measurements. The macro used for ATG8 puncta analysis was ideal in our case also because we looked at relative differences among different treatments that were quite substantial. We also provided biochemical evidence supporting these analyses.

2. Authors conclude that PexRD54 imitates carbon starvation conditions that trigger a specific type of autophagic response. However, no other types of autophagy-inducing conditions were compared to PexRD54 mode of actions.

We agree that we did not expensively looked into other autophagy inducing conditions and revised the text accordingly (Please see responses to editor/s comment point 3). However, we also looked at Joka2-induced autophagy pathway (aggrephagy) in Figure 6.

3. PexRD54 is not only diverting autophagy but is also being degraded by it. To better clarify the dynamics of this balance, it would be beneficial to add to the current data a comparison of differences in the rate of autophagosome delivery to the vacuole under carbon starvation-induced autophagy and autophagy induced by PexRD54.

This is a good point. We will investigate this in a follow up manuscript.